# Efficient Stochastic Gradient Hard Thresholding

**Pan Zhou**[*]        **Xiao-Tong Yuan**[†]        **Jiashi Feng**[*]
[*] Learning & Vision Lab, National University of Singapore, Singapore
[†] B-DAT Lab, Nanjing University of Information Science & Technology, Nanjing, China
pzhou@u.nus.edu        xtyuan@nuist.edu.cn        elefjia@nus.edu.sg

## Abstract

Stochastic gradient hard thresholding methods have recently been shown to work favorably in solving large-scale empirical risk minimization problems under sparsity or rank constraint. Despite the improved iteration complexity over full gradient methods, the gradient evaluation and hard thresholding complexity of the existing stochastic algorithms usually scales linearly with data size, which could still be expensive when data is huge and the hard thresholding step could be as expensive as singular value decomposition in rank-constrained problems. To address these deficiencies, we propose an efficient hybrid stochastic gradient hard thresholding (HSG-HT) method that can be provably shown to have sample-size-independent gradient evaluation and hard thresholding complexity bounds. Specifically, we prove that the stochastic gradient evaluation complexity of HSG-HT scales linearly with inverse of sub-optimality and its hard thresholding complexity scales logarithmically. By applying the heavy ball acceleration technique, we further propose an accelerated variant of HSG-HT which can be shown to have improved factor dependence on restricted condition number in the quadratic case. Numerical results confirm our theoretical affirmation and demonstrate the computational efficiency of the proposed methods.

## 1 Introduction

We consider the following sparsity- or rank-constrained *finite-sum* minimization problems which are widely applied in high-dimensional statistical estimation:

$$\min_{\boldsymbol{x}} f(\boldsymbol{x}) := \frac{1}{n} \sum\nolimits_{i=1}^{n} f_i(\boldsymbol{x}), \quad \text{s.t. } \|\boldsymbol{x}\|_0 \leq k \quad \text{or} \quad \mathsf{rank}\,(\boldsymbol{x}) \leq k, \qquad (1)$$

where each individual loss $f_i(\boldsymbol{x})$ is associated with the $i$-th sample, $\|\boldsymbol{x}\|_0$ denotes the number of nonzero entries in $\boldsymbol{x}$ as a vector variable, $\mathsf{rank}\,(\boldsymbol{x})$ denotes the rank of $\boldsymbol{x}$ as a matrix variable, and $k$ represents the sparsity/low-rankness level. Such a formulation encapsulates several important problems, including $\ell_0$-constrained linear/logistic regression [1, 2, 3], sparse graphical model learning [4], and low-rank multivariate and multi-task regression [5, 6, 7, 8, 9], to name a few.

We are particularly interested in gradient hard thresholding methods [10, 11, 12, 13] which are popular and effective for solving problem (1). The common theme of this class of methods is to iterate between gradient descent and hard thresholding to maintain sparsity/low-rankness of solution while minimizing the loss function. In our problem setting, a plain gradient hard thresholding iteration is given by $\boldsymbol{x}^{t+1} = \Phi_k(\boldsymbol{x}^t - \eta \nabla f(\boldsymbol{x}^t))$, where $\Phi_k(\cdot)$, as defined in Section 2, denotes the hard thresholding operation that preserves the top $k$ entries (in magnitude) of a vector or produces an optimal rank-$k$ approximation to a matrix via singular value decomposition (SVD). When considering gradient hard thresholding methods, two main sources of computational complexity are at play: the gradient evaluation complexity and the hard thresholding complexity. As the per-iteration hard thresholding can be as expensive as SVD in rank-constrained problems, our goal is to develop methods that iterate and converge quickly while using a minimal number of hard thresholding operations.

Table 1: Comparison of different hard thresholding algorithms for sparsity- and rank-constrained problem (1). Both computational complexity and statistical error are evaluated w.r.t. the estimation error $\|\tilde{x} - x^*\|$ between the $k$-sparse/rank estimator $\tilde{x}$ and the $k^*$-sparse/rank optimum $x^*$. Both $\kappa_s$ and $\kappa_{\widehat{s}}$ denote the restricted condition numbers with $s = 2k + k^*$ and $\widehat{s} = 3k + k^*$. $\widetilde{\mathcal{I}} = \mathsf{supp}(\Phi_{2k}(\nabla f(x^*))) \cup \mathsf{supp}(x^*)$ and $\widehat{\mathcal{I}} = \mathsf{supp}(\Phi_{3k}(\nabla f(x^*))) \cup \mathsf{supp}(x^*)$ are two support sets. The results of AHSG-HT are established for quadratic loss functions.

| | Restriction on $\kappa_s$ | Required value of $k$ | Computational Complexity | | Statistical Error on Sparsity-constrained Problem[1] |
| --- | --- | --- | --- | --- | --- |
| | | | #IFO | #Hard Thresholding | |
| FG-HT [12, 13] | — | $\Omega(\kappa_s^2 k^*)$ | $\mathcal{O}\big(n\kappa_s \log\big(\frac{1}{\epsilon}\big)\big)$ | $\mathcal{O}\big(\kappa_s \log\big(\frac{1}{\epsilon}\big)\big)$ | $\mathcal{O}\big(\sqrt{k+k^*}\|\nabla f(x^*)\|_\infty\big)$ |
| SG-HT [20] | $\leq \frac{4}{3}$ | $\Omega(\kappa_s^2 k^*)$ | $\mathcal{O}\big(\kappa_s \log\big(\frac{1}{\epsilon}\big)\big)$ | $\mathcal{O}\big(\kappa_s \log\big(\frac{1}{\epsilon}\big)\big)$ | $\mathcal{O}\big(\frac{1}{n}\sum_{i=1}^n \|\nabla f_i(x^*)\|_2\big)$ |
| SVRG-HT [21] | — | $\Omega(\kappa_s^2 k^*)$ | $\mathcal{O}\big((n+\kappa_s)\log\big(\frac{1}{\epsilon}\big)\big)$ | $\mathcal{O}\big((n+\kappa_s)\log\big(\frac{1}{\epsilon}\big)\big)$ | $\mathcal{O}\big(\sqrt{s}\|\nabla f(x^*)\|_\infty + \|\nabla_{\widetilde{\mathcal{I}}} f(x^*)\|_2\big)$ |
| HSG-HT | — | $\Omega(\kappa_s^2 k^*)$ | $\mathcal{O}\big(\frac{\kappa_s}{\epsilon}\big)$ | $\mathcal{O}\big(\kappa_s \log\big(\frac{1}{\epsilon}\big)\big)$ | $\mathcal{O}\big(\|\nabla_{\widetilde{\mathcal{I}}} f(x^*)\|_2\big)$ |
| AHSG-HT | — | $\Omega(\kappa_{\widehat{s}} k^*)$ | $\mathcal{O}\big(\frac{\sqrt{\kappa_{\widehat{s}}}}{\epsilon}\big)$ | $\mathcal{O}\big(\sqrt{\kappa_{\widehat{s}}}\log\big(\frac{1}{\epsilon}\big)\big)$ | $\mathcal{O}\big(\|\nabla_{\widehat{\mathcal{I}}} f(x^*)\|_2\big)$ |

[1] For general rank-constrained problem, the statistic error is not explicitly provided in FG-HT, SG-HT and SVRG-HT while is given in our Theorem 1 for HSG-HT and Theorem 3 for AHSG-HT.

**Full gradient hard thresholding.** The plain form of full gradient hard thresholding (FG-HT) algorithm has been extensively studied in compressed sensing and sparse learning [10, 12, 13, 14]. At each iteration, FG-HT first updates the variable $x$ by using full gradient descent and then performs hard thresholding on the updated variable. Theoretical results show that FG-HT converges linearly towards a proper nominal solution with high estimation accuracy [12, 13, 15]. Besides, compared with the algorithms adopting $\ell_1$- or nuclear-norm convex relaxation (*e.g.*, [16, 17, 18, 19]), directly solving problem (1) via FG-HT often exhibits similar accuracy guarantee but is more computationally efficient. However, despite these desirable properties, FG-HT needs to compute the full gradient at each iteration which can be expensive in large-scale problems. If the restricted condition number is $\kappa_s$, then $\mathcal{O}\big(\kappa_s \log\big(\frac{1}{\epsilon}\big)\big)$ iterations are needed to attain an $\epsilon$-suboptimal solution (up to a statistical error), and thus the sample-wise gradient evaluation complexity, or incremental first order oracle (IFO, see Definition 1), is $\mathcal{O}\big(n\kappa_s \log\big(\frac{1}{\epsilon}\big)\big)$ which scales linearly with $n\kappa_s$.

**Stochastic gradient hard thresholding.** To improve computational efficiency, stochastic hard thresholding algorithms [20, 21, 22] have recently been developed via leveraging the finite-sum structure of problem (1). For instance, Nguyen *et al.* [20] proposed a stochastic gradient hard thresholding (SG-HT) algorithm for solving problem (1). At each iteration, SG-HT only evaluates gradient of one (or a mini-batch) randomly selected sample for variable update and hard thresholding. It was shown that the IFO complexity and hard thresholding complexity of SG-HT are both $\mathcal{O}\big(\kappa_s \log\big(\frac{1}{\epsilon}\big)\big)$ which is independent on $n$. However, SG-HT can only be shown to converge to a sub-optimal statistical estimation accuracy (see Table 1) which is inferior to that of the full-gradient methods. Another limitation of SG-HT is that it requires the restricted condition number $\kappa_s$ to be not larger than $4/3$ which is hard to meet in realistic high-dimensional sparse estimation problems such as sparse linear regression [13]. To overcome these issues, the stochastic variance reduced gradient hard thresholding (SVRG-HT) algorithm [21, 22] was developed as an adaptation of SVRG [23] to problem (1). Benefiting from the variance reduced technique, SVRG-HT can converge more stably and efficiently while having better estimation accuracy than SG-HT. Also different from SG-HT, the convergence analysis for SVRG-HT allows arbitrary bounded restricted condition number. As shown in Table 1, both the IFO complexity and hard thresholding complexity of SVRG-HT are $\mathcal{O}\big((n + \kappa_s) \log\big(\frac{1}{\epsilon}\big)\big)$. Although the IFO complexity of SVRG-HT substantially improves over FG-HT, the overall complexity still scale linearly with respect to the sample size $n$. Therefore, when the data-scale is huge (e.g., $n \gg \kappa_s$) and the per-iteration hard thresholding operation is expensive, SVRG-HT could still be computationally inefficient in practice. Later, Chen *et al.* [24] proposed a stochastic variance-reduced block coordinate descent algorithm. But its overall complexity still scale linearly with respect to the sample size and thus it faces the same challenge as SVRG-HT in computation.

**Overview of our approach.** The method we propose can be viewed as a simple yet efficient extension of the hybrid stochastic gradient descent (HSGD) method [25, 26, 27] from unconstrained finite-sum minimization to the cardinality-constrained finite-sum problem (1). The core idea of HSGD is to iteratively sample an evolving mini-batch of terms in the finite-sum for gradient estimation. This style of incremental gradient method has been shown, both in theory and practice, to bridge smoothly the gap between deterministic and stochastic gradient methods [26]. Inspired by the success of

HSGD, we propose the hybrid stochastic gradient hard thresholding (HSG-HT) method which has the following variable update form:

$$\boldsymbol{x}^{t+1} = \Phi_k \left(\boldsymbol{x}^t - \eta \boldsymbol{g}^t\right), \text{with } \boldsymbol{g}^t = \frac{1}{s_t} \sum\nolimits_{i_t \in \mathcal{S}_t} \nabla f_{i_t}(\boldsymbol{x}^t),$$

where $\eta$ is the learning rate and $\mathcal{S}_t$ is the set of $s_t$ selected samples. In early stage of iterations, HSG-HT selects a few samples to estimate the full gradient; and along with more iterations, $s_t$ increases, giving more accurate full gradient estimation. Such a mechanism allows it to enjoy the merits of both SG-HT and FG-HT, *i.e.* the low iteration complexity of SG-HT and the steady convergence rate of FG-HT with constant learning rate $\eta$. Given a $k^*$-sparse/low-rank target solution $\boldsymbol{x}^*$, for objective function with restricted condition number $\kappa_s$ and $s = 2k + k^*$, we show that $\mathcal{O}\left(\frac{\kappa_s}{\epsilon}\right)$ rounds of IFO update and $\mathcal{O}\left(\kappa_s \log \left(\frac{1}{\epsilon}\right)\right)$ steps of hard thresholding operation are sufficient for HSG-HT to find $\tilde{\boldsymbol{x}}$ such that $\|\tilde{\boldsymbol{x}} - \boldsymbol{x}^*\|^2 \leq \epsilon + \mathcal{O}\left(s\|\nabla f(\boldsymbol{x}^*)\|_\infty^2\right)$. In this way, HSG-HT exhibits sample-size-independent IFO and hard thresholding complexity. Another attractiveness of HSG-HT is that it can be readily accelerated via applying the heavy ball acceleration technique [28, 29, 30]. To this end, we modify the iteration of HSG-HT by adding a small momentum $\nu(\boldsymbol{x}^t - \boldsymbol{x}^{t-1})$ for some $\nu > 0$ to the gradient descent step:

$$\boldsymbol{x}^{t+1} = \Phi_k \left(\boldsymbol{x}^t - \eta \boldsymbol{g}^t + \nu(\boldsymbol{x}^t - \boldsymbol{x}^{t-1})\right).$$

We call the above modified version as accelerated HSG-HT (AHSG-HT). For quadratic problems, we prove that such a simple momentum strategy boosts the IFO complexity of HSG-HT to $\mathcal{O}\left(\frac{\sqrt{\kappa_{\widehat{s}}}}{\epsilon}\right)$, and the hard thresholding complexity to $\mathcal{O}\left(\sqrt{\kappa_{\widehat{s}}} \log \left(\frac{1}{\epsilon}\right)\right)$, where $\widehat{s} = 3k + k^*$. To the best of our knowledge, AHSG-HT is the first momentum based algorithm that can be provably shown to have such an improved complexity for stochastic gradient hard thresholding.

**Highlight of results and contribution.** Table 1 summarizes our main results on computational complexity and statistical estimation accuracy of HSG-HT and AHSG-HT, along with the results for the above mentioned state-of-the-art gradient hard thresholding algorithms. From this table we can observe that our methods have several theoretical advantages over the considered prior methods, which are highlighted in the next few paragraphs.

*On sparsity/low-rankness level constraint condition.* AHSG-HT in the quadratic case substantially improves the bounding condition on the sparsity/low-rankness level $k$: it only requires $k = \Omega(\kappa_{\widehat{s}} k^*)$, while the other considered algorithms with optimal statistical estimation accuracy all require $k = \Omega(\kappa_s^2 k^*)$. Moreover, both HSG-HT and AHSG-HT get rid of the restrictive condition $\kappa_s \leq 4/3$ required in SG-HT.

*On statistical estimation accuracy.* For sparsity-constrained problem, the statistical estimation accuracy of HSG-HT is comparable to that in FG-HT and is better than those in SVRG and SG-HT, as $\|\nabla_{\widetilde{\mathcal{I}}} f(\boldsymbol{x}^*)\|_2$ in HSG-HT is usually superior to the error $\sqrt{s}\|\nabla f(\boldsymbol{x}^*)\|_\infty$ in SVRG-HT and is much smaller than the one $\frac{1}{n} \sum_{i=1}^{n} \|\nabla f_i(\boldsymbol{x}^*)\|_2$ in SG-HT. AHSG-HT has even smaller estimation error than HSG-HT since it allows smaller sparsity/low-rankness level $k = \Omega(\kappa_{\widehat{s}} k^*)$ and thus a smaller cardinality of the support set $\widehat{\mathcal{I}}$, especially when the restrictive condition number is sensitive to $k$.

*On computational complexity.* Both HSG-HT and AHSG-HT enjoy sample-size-independent IFO and hard thresholding complexity. To compare the IFO complexity, our methods will be cheaper than FG-HT and SVRG-HT when $n$ dominates $\frac{1}{\epsilon}$. This suggests that HSG-HT and AHSG-HT are more suitable for handling large-scale data. SG-HT has the lowest IFO complexity, which however is obtained at the price of severely deteriorated statistical estimation accuracy. In terms of hard thresholding complexity, AHSG-HT is the best one and HSG-HT matches FG-HT and SG-HT.

Last but not least, we highlight that AHSG-HT, to our best knowledge, for the first time provides improved convergence guarantees for momentum based stochastic gradient hard thresholding methods. While in convex problems the momentum based methods such as heavy ball and Nesterov's methods have long been known to work favorably for accelerating full/stochastic gradient methods [28, 31, 32, 33], it still remains largely unknown if it is possible to accelerate gradient hard thresholding methods for solving the non-convex finite-sum problem (1). There is a recent attempt at understanding a Nesterov's momentum full gradient hard thresholding method [34]. Although showing to attain improved rate of convergence under certain conditions, the iteration complexity bound established in [34] still does not exhibit better dependence on restricted condition number than the plain FG-HT. In contrast, at least in the quadratic case, AHSG-HT can be shown to have improved dependence on condition number than the existing gradient hard thresholding methods.

## 2 Preliminaries

Throughout this paper, we use $\|\boldsymbol{x}\|$ to denote the Euclidean norm for vector $\boldsymbol{x} \in \mathbb{R}^d$ and the Frobenius norm for matrix $\boldsymbol{x} \in \mathbb{R}^{d_1 \times d_2}$. $\|\boldsymbol{x}\|_\infty$ denotes the largest absolute entry in $\boldsymbol{x}$. The hard thresholding operation $\Phi_k(\boldsymbol{x})$ preserves the $k$ largest entries of $\boldsymbol{x}$ in magnitude for vector $\boldsymbol{x}$, and for matrix $\boldsymbol{x}$ it only preserves the top $k$-top singular values. Namely, $\Phi_k(\boldsymbol{x}) = \boldsymbol{H}_k \boldsymbol{\Sigma}_k \boldsymbol{V}_k^T$, where $\boldsymbol{H}_k$ and $\boldsymbol{V}_k$ are respectively the top-$k$ left and right singular vectors of $\boldsymbol{x}$, $\boldsymbol{\Sigma}_k$ is the diagonal matrix of the top-$k$ singular values of $\boldsymbol{x}$. We use $\mathsf{supp}(\boldsymbol{x})$ to denote the support set of $\boldsymbol{x}$. Specifically, for vector $\boldsymbol{x}$, $\mathsf{supp}(\boldsymbol{x})$ indexes its nonzero entries; and for matrix $\boldsymbol{x} \in \mathbb{R}^{d_1 \times d_2}$, it indexes the subspace $\boldsymbol{U}$ that is a set of singular vectors spanning the column space of $\boldsymbol{x}$. For vector variable $\boldsymbol{x}$, $\nabla_\mathcal{I} f(\boldsymbol{x})$ preserves the entries in $\nabla f(\boldsymbol{x})$ indexed by the support set $\mathcal{I}$ and sets the remaining entries to be zero; while for matrix variable $\boldsymbol{x}$, $\nabla_\mathcal{I} f(\boldsymbol{x})$ with $\mathcal{I} = \mathcal{I}_1 \cup \mathcal{I}_2$ projects $\nabla f(\boldsymbol{x})$ into the subspace indexed by $\mathcal{I}_1 \cup \mathcal{I}_2$, namely $\nabla_\mathcal{I} f(\boldsymbol{x}) = (\boldsymbol{U}_1 \boldsymbol{U}_1^T + \boldsymbol{U}_2 \boldsymbol{U}_2^T - \boldsymbol{U}_1 \boldsymbol{U}_1^T \boldsymbol{U}_2 \boldsymbol{U}_2^T) \nabla f(\boldsymbol{x})$, where $\boldsymbol{U}_1$ and $\boldsymbol{U}_2$ respectively span the subspaces indexed by $\mathcal{I}_1$ and $\mathcal{I}_2$.

We assume the objective function in (1) to have restricted strong convexity (RSC) and restricted strong smoothness (RSS). For both sparsity- and rank-constrained problems, the RSC and RSS conditions are commonly used in analyzing hard thresholding algorithms [12, 13, 21, 22, 20].

**Assumption 1** (Restricted strong convexity condition, RSC). *A differentiable function $f(\boldsymbol{x})$ is restricted $\rho_s$-strongly convex with parameter $s$ if there exists a generic constant $\rho_s > 0$ such that for any $\boldsymbol{x}$, $\boldsymbol{x}'$ with $\|\boldsymbol{x} - \boldsymbol{x}'\|_0 \le s$ or $\mathsf{rank}\,(\boldsymbol{x} - \boldsymbol{x}') \le s$,*

$$f(\boldsymbol{x}) - f(\boldsymbol{x}') - \langle \nabla f(\boldsymbol{x}'), \boldsymbol{x} - \boldsymbol{x}' \rangle \ge \frac{\rho_s}{2} \|\boldsymbol{x} - \boldsymbol{x}'\|^2.$$

**Assumption 2** (Restricted strong smoothness condition, RSS). *For each $f_i(\boldsymbol{x})$, it is said to be restricted $\ell_s$-strongly smooth with parameter $s$ if there exists a generic constant $\ell_s > 0$ such that for any $\boldsymbol{x}$, $\boldsymbol{x}'$ with $\|\boldsymbol{x} - \boldsymbol{x}'\|_0 \le s$ or $\mathsf{rank}\,(\boldsymbol{x} - \boldsymbol{x}') \le s$,*

$$f_i(\boldsymbol{x}) - f_i(\boldsymbol{x}') - \langle \nabla f_i(\boldsymbol{x}'), \boldsymbol{x} - \boldsymbol{x}' \rangle \le \frac{\ell_s}{2} \|\boldsymbol{x} - \boldsymbol{x}'\|^2.$$

We also need to impose the following boundness assumption on the variance of stochastic gradient.

**Assumption 3** (Bounded stochastic gradient variance). *For any $\boldsymbol{x}$ and each loss $f_i(\boldsymbol{x})$, the distance between $\nabla f_i(\boldsymbol{x})$ and the full gradient $\nabla f(\boldsymbol{x})$ is upper bounded as $\max_i \|\nabla f_i(\boldsymbol{x}) - \nabla f(\boldsymbol{x})\| \le B$.*

Similar to [21, 23, 35, 36], the incremental first order oracle (IFO) complexity is adopted as the computational complexity metric for solving finite-sum problem (1). In high-dimensional sparse learning and low-rank matrix recovery problems, the per-iteration hard thresholding operation can be equally time-consuming or even more expensive than gradient evaluation. For instance, in rank-constrained problems, hard thresholding operation can be as expensive as top-$k$ SVD for a matrix. Therefore we also need to take the computational complexity of hard thresholding into our account.

**Definition 1** (IFO and Hard Thresholding Complexity). *For $f(\boldsymbol{x})$ in problem (1), an IFO takes an index $i \in [n]$ and a point $\boldsymbol{x}$, and returns the pair $(f_i(\boldsymbol{x}), \nabla f_i(\boldsymbol{x}))$. In a hard thresholding operation, we feed $\boldsymbol{x}$ into $\Phi_k(\cdot)$ and obtain the output $\Phi_k(\boldsymbol{x})$.*

The IFO and hard thresholding complexity as a whole can more comprehensively reflect the overall computational performance of a first-order hard thresholding algorithm, as objective value, gradient evaluation and hard thresholding operation usually dominate the per-iteration computation.

## 3 Hybrid Stochastic Gradient Hard Thresholding

In this section, we first introduce the Hybrid Stochastic Gradient Hard Thresholding (HSG-HT) algorithm and then analyze its convergence performance for sparsity- and rank-constrained problems.

### 3.1 The HSG-HT Algorithm

The HSG-HT algorithm is outlined in Algorithm 1. At the $t$-th iteration, it first uniformly randomly selects $s_t$ samples $\mathcal{S}_t$ from all data and evaluates the approximated gradient $\boldsymbol{g}^t = \frac{1}{s_t} \sum_{i_t \in \mathcal{S}_t} \nabla f_{i_t}(\boldsymbol{x}^t)$.

**Algorithm 1:** (Accelerated) Hybrid Stochastic Gradient Hard Thresholding

---

**Input** : Initial point $\boldsymbol{x}^0$, sample index set $\mathcal{S} = \{1, \cdots, n\}$, learning rate $\eta$, momentum strength $\nu$,
mini-batch sizes $\{s_t\}$.

**for** $t = 1, 2, ..., T - 1$ **do**

   Uniformly randomly select $s_t$ samples $\mathcal{S}_t$ from $\mathcal{S}$
   Compute the approximate gradient $\boldsymbol{g}^t = \frac{1}{s_t} \sum_{i_t \in \mathcal{S}_t} \nabla f_{i_t}(\boldsymbol{x}^t)$

   Update $\boldsymbol{x}^{t+1}$ using either of the following two options:
     **(O1)** $\boldsymbol{x}^{t+1} = \Phi_k\left(\boldsymbol{x}^t - \eta \boldsymbol{g}^t\right)$; /* for plain HSG-HT */
     **(O2)** $\boldsymbol{x}^{t+1} = \Phi_k\left(\boldsymbol{x}^t - \eta \boldsymbol{g}^t + \nu(\boldsymbol{x}^t - \boldsymbol{x}^{t-1})\right)$; /* for accelerated HSG-HT */

**end**

**Output** : $\boldsymbol{x}^T$.

---

Then, there are two options for variable update. The first option is to update $\boldsymbol{x}^{t+1}$ with a standard local descent step along $\boldsymbol{g}^t$ followed by a hard thresholding step, giving the plain update procedure $\boldsymbol{x}^{t+1} = \Phi_k\left(\boldsymbol{x}^t - \eta \boldsymbol{g}^t\right)$ in option **O1**. The other option **O2** is to update $\boldsymbol{x}^{t+1}$ based on a momentum formulation $\boldsymbol{x}^{t+1} = \Phi_k\left(\boldsymbol{x}^t - \eta \boldsymbol{g}^t + \nu(\boldsymbol{x}^t - \boldsymbol{x}^{t-1})\right)$, leading to an accelerated variant of HSG-HT. The plain update **O1** is actually a special case of the momentum based update in **O2** with strength $\nu = 0$. In early stage of iteration when the mini-batch size $s_t$ is relatively small, HSG-HT performs more like SG-HT with low per-iteration gradient evaluation cost. Along with more iterations, $s_t$ increases and HSG-HT performs like full gradient hard thresholding methods. Next, we analyze the parameter estimation accuracy and the objective value convergence of HSG-HT. The analysis of the accelerated version will be presented in Section 4.

## 3.2  Statistical Estimation Analysis

We first analyze the parameter estimation performance of HSG-HT by characterizing the distance between the output of Algorithm 1 and the optimum $\boldsymbol{x}^*$. Such an analysis is helpful in understanding the convergence behavior and statistical estimation accuracy of the computed solution. We summarize the main result for both sparsity- and rank-constrained problems in Theorem 1.

**Theorem 1.** *Suppose the objective function $f(\boldsymbol{x})$ is $\rho_s$-strongly convex and each individual $f_i(\boldsymbol{x})$ is $\ell_s$-strongly smooth with parameter $s = 2k + k^*$. Let $\kappa_s = \frac{\ell_s}{\rho_s}$ and $\alpha = 1 + \frac{2\sqrt{k^*}}{\sqrt{k - k^*}}$. Assume the sparsity/low-rankness level $k \geq \left(1 + 712\kappa_s^2\right) k^*$. Set the learning rate $\eta = \frac{1}{6\ell_s}$ and the mini-batch size $s_t = \frac{\tau}{\omega^t}$ with $\omega = 1 - \frac{1}{480\kappa_s}$ and $\tau \geq \frac{40\alpha B}{3\rho_s \ell_s \|\boldsymbol{x}^0 - \boldsymbol{x}^*\|^2}$. Then the output $\boldsymbol{x}^T$ of HSG-HT satisfies*

$$\mathbb{E}\|\boldsymbol{x}^T - \boldsymbol{x}^*\| \leq \left(1 - \frac{1}{480\kappa_s}\right)^{T/2}\|\boldsymbol{x}^0 - \boldsymbol{x}^*\| + \frac{\sqrt{\alpha}}{\ell_s\sqrt{12(1 - \beta)}}\|\nabla_{\widetilde{\mathcal{I}}} f(\boldsymbol{x}^*)\|,$$

*where $\beta = \alpha\left(1 - \frac{1}{12\kappa_s}\right)$, $\widetilde{\mathcal{I}} = \mathsf{supp}(\Phi_{2k}(\nabla f(\boldsymbol{x}^*))) \cup \mathsf{supp}(\boldsymbol{x}^*)$, and $T$ is the number of iterations.*

A proof of Theorem 1 is given in Appendix B.1. Theorem 1 shows that for both sparsity- and rank-constrained problem, if using sparsity/low-rankness level $k = \Omega(\kappa_s^2 k^*)$ and gradually expanding the mini-batch size at an exponential rate of $\frac{1}{\omega}$ with $\omega = 1 - \frac{1}{480\kappa_s}$, then in expectation the sequence $\{\boldsymbol{x}^t\}$ generated by HSG-HT converges linearly towards $\boldsymbol{x}^*$ at the rate of $\left(1 - \frac{1}{480\kappa_s}\right)^{\frac{1}{2}}$. This indicates that HSG-HT enjoys a similar fast and steady convergence rate just like the deterministic FG-HT [13]. As the condition number $\kappa_s = \ell_s/\rho_s$ is usually large in realistic problems, the exponential rate $\frac{1}{\omega}$ is actually only a slightly larger than one. This means even a moderate-scale dataset will allow HSGD-HT to iterate sufficiently for decreasing the loss, as illustrated in Figure 1 and 2 in Section 5.

One can also observe that the estimation error of $\mathbb{E}\|\boldsymbol{x}^t - \boldsymbol{x}^*\|$ is controlled by the multiplier of $\|\nabla_{\widetilde{\mathcal{I}}} f(\boldsymbol{x}^*)\|$ which usually represents the statistical error of model. For sparsity-constrained problem, such a statistical error bound matches that established in FG-HT [13], and is usually better than the error bound $\mathcal{O}\left(\sqrt{s}\|\nabla f(\boldsymbol{x}^*)\|_\infty + \|\nabla_{\widetilde{\mathcal{I}}} f(\boldsymbol{x}^*)\|_2\right)$ with $s = 2k + k^*$ in SVRG-HT [21] since $\|\nabla_{\widetilde{\mathcal{I}}} f(\boldsymbol{x}^*)\|_2 \leq \sqrt{s}\|\nabla f(\boldsymbol{x}^*)\|_\infty$. Compared with the error $\mathcal{O}\left(\frac{1}{n}\sum_{i=1}^n \|\nabla f_i(\boldsymbol{x}^*)\|_2\right)$ for SG-HT [20], the error term of HSG-HT is significantly smaller. This is because the magnitude $\|\nabla f(\boldsymbol{x}^*)\|_2$ of the full gradient is usually small when sample size is large, while the individual (or small mini-batch) gradient norm $\|\nabla f_i(\boldsymbol{x}^*)\|_2$ could still have relatively large magnitude. For

example, in sparse linear regression problems the difference could be as significant as $\mathcal{O}(\sqrt{\log(d)/n})$ (in HSG-HT) versus $\mathcal{O}(\sqrt{\log(d)})$ (in SG-HT). Notice, for the general rank-constrained problem, FG-HT, SG-HT and SVRG-HT do not explicitly provide the statistical error as given by HSG-HT. Indeed, SG-HT and SVRG-HT only considered a low-rank matrix linear model which is a special case of the general rank-constrained problem (1). Moreover, to guarantee convergence, SG-HT requires the restrictive condition $\kappa_s \leq 4/3$, while our analysis removes such a condition and allows for an arbitrarily large $\kappa_s$ as long as it is bounded.

Based on Theorem 1, we can derive the IFO and hard thresholding complexity of HSG-HT for problem (1) in Corollary 1 with proof in Appendix B.2. For fairness, here we follow the convention in [13, 20, 21, 22] to use $\mathbb{E}\|\boldsymbol{x} - \boldsymbol{x}^*\| \leq \sqrt{\epsilon} +$ statistical error as the measure of $\epsilon$-suboptimality.

**Corollary 1.** *Suppose the conditions in Theorem 1 hold. To achieve* $\mathbb{E}\|\boldsymbol{x}^T - \boldsymbol{x}^*\| \leq \sqrt{\epsilon} + \frac{\sqrt{\alpha}\|\nabla_{\widetilde{\mathcal{I}}}f(\boldsymbol{x}^*)\|}{\ell_s\sqrt{12(1-\beta)}}$, *the IFO complexity of HSG-HT in Algorithm 1 is* $\mathcal{O}\left(\frac{\kappa_s}{\epsilon}\right)$ *and the hard threshold-ing complexity is* $\mathcal{O}\left(\kappa_s \log\left(\frac{1}{\epsilon}\right)\right)$.

Compared with FG-HT [13] and SVRG-HT [21, 22] whose IFO complexity are $\mathcal{O}\left(n\kappa_s \log\left(\frac{1}{\epsilon}\right)\right)$ and $\mathcal{O}\left((n + \kappa_s) \log\left(\frac{1}{\epsilon}\right)\right)$ respectively, HSG-HT is more computationally efficient in IFO than FG-HT and SVRG-HT when sample size $n$ dominates $\frac{1}{\epsilon}$. This is usually the case when the data scale is huge while the desired accuracy $\epsilon$ is moderately small. Concerning the hard thresholding complexity, HSG-HT shares the same complexity $\mathcal{O}\left(\kappa_s \log\left(\frac{1}{\epsilon}\right)\right)$ with FG-HT, which is considerably cheaper than the $\mathcal{O}\left((n + \kappa_s) \log\left(\frac{1}{\epsilon}\right)\right)$ hard thresholding complexity of SVRG-HT when data scale is large. Overall, HSG-HT is able to achieve better trade-off between IFO and hard thresholding complexity than FG-HT and SVRG-HT when $n$ is much larger than $\frac{1}{\epsilon}$ in large-scale learning problems.

## 3.3 Convergence Analysis

For sparsity-constrained problem, we further investigate the convergence behavior of HSG-HT in terms of the objective value $f(\boldsymbol{x})$ towards the optimal loss $f(\boldsymbol{x}^*)$. The main result is summarized in the following theorem, whose proof is deferred to Appendix B.3.

**Theorem 2.** *Suppose $f(\boldsymbol{x})$ is $\rho_s$-strongly convex and each individual component $f_i(\boldsymbol{x})$ is $\ell_s$-strongly smooth with parameter $s = 2k + k^*$. Let $\kappa_s = \frac{\ell_s}{\rho_s}$ and the sparsity level $k \geq \left(1 + 64\kappa_s^2\right)k^*$. By setting the learning rate $\eta = \frac{1}{2\ell_s}$ and the mini-batch size $s_t = \frac{\tau}{\omega^t}$ with $\omega = 1 - \frac{1}{16\kappa_s}$ and $\tau \geq \frac{148B\kappa_s^2}{\rho_s[f(\boldsymbol{x}^0)-f(\boldsymbol{x}^*)]}$, then for sparsity-constrained problem, the output $\boldsymbol{x}^T$ of Algorithm 1 satisfies*

$$\mathbb{E}[f(\boldsymbol{x}^T) - f(\boldsymbol{x}^*)] \leq \left(1 - \frac{1}{16\kappa_s}\right)^T [f(\boldsymbol{x}^0) - f(\boldsymbol{x}^*)].$$

Theorem 2 shows that for sparsity-constrained problem, HSG-HT in expectation also enjoys linear convergence in terms of the objective value by gradually exponentially expanding the mini-batch size. The result in Theorem 2 also implies that the expected value of $f(\boldsymbol{x}^t)$ can be arbitrarily close to the $k^*$-sparse target value $f(\boldsymbol{x}^*)$ as long as the iteration number is sufficiently large. This property is important, since in realistic problems, such as classification or regression problems, if $f(\boldsymbol{x})$ is more close to the optimum $f(\boldsymbol{x}^*)$, then the prediction result can be better. FG-HT [13] also enjoys such a good property. In contrast, for SVRG-HT [21], the convergence bound is known to be $\mathbb{E}[f(\boldsymbol{x}^t) - f(\boldsymbol{x}^*)] \leq \mathcal{O}\left(\zeta^t + \sqrt{s}\|\nabla f(\boldsymbol{x}^*)\|_\infty\right)$ for some shrinkage rate $\zeta \in (0, 1)$. That result is inferior to ours due to the presence of a non-vanishing statistical barrier term $\sqrt{s}\|\nabla f(\boldsymbol{x}^*)\|_\infty$.

# 4 Acceleration via Heavy-Ball Method

In this section, we show that HSG-HT can be effectively accelerated by applying the heavy ball technique [28, 29]. As proposed in the option **O2** in Algorithm 1, the idea is to use the integration of the estimated gradient $\boldsymbol{g}^t$ and a small momentum $\nu(\boldsymbol{x}^t - \boldsymbol{x}^{t-1})$ to modify the update as $\boldsymbol{x}^{t+1} = \Phi_k\left(\boldsymbol{x}^t - \eta\boldsymbol{g}^t + \nu(\boldsymbol{x}^t - \boldsymbol{x}^{t-1})\right)$. The following result confirms that such an accelerated variant, *i.e.* AHSG-HT, can significantly improve the efficiency of HSG-HT for quadratic loss functions. A proof of this result can be found in Appendix C.1.

**Theorem 3.** *Suppose the objective function $f(\boldsymbol{x})$ is quadratic and it satisfies the RSC and RSS conditions with parameter $\widehat{s} = 3k + k^*$. Let $\kappa_{\widehat{s}} = \frac{\ell_{\widehat{s}}}{\rho_{\widehat{s}}}$. Assume the sparsity/low-rankness level $k \geq (1 + 16\kappa_{\widehat{s}}) k^*$. Set the learning rate $\eta = \frac{4}{(\sqrt{\ell_{\widehat{s}}} + \sqrt{\rho_{\widehat{s}}})^2}$, the mini-batch size $s_t = \frac{\tau}{\omega^t}$ where $\omega = (1 - \frac{1}{18\sqrt{\kappa_{\widehat{s}}}})^2$ and $\tau \geq \frac{81B\kappa_{\widehat{s}}}{4(\sqrt{\rho_{\widehat{s}}} + \sqrt{\ell_{\widehat{s}}})^4 \|\boldsymbol{x}^0 - \boldsymbol{x}^*\|^2}$, the momentum parameter $\nu = \left(\frac{\sqrt{\kappa_{\widehat{s}}} - 1}{\sqrt{\kappa_{\widehat{s}}} + 1}\right)^2$. Then the output $\boldsymbol{x}^T$ of AHSG-HT in Algorithm 1 satisfies*

$$\mathbb{E}\|\boldsymbol{x}^T - \boldsymbol{x}^*\| \leq 2\left(1 - \frac{1}{18\sqrt{\kappa_s}}\right)^T \|\boldsymbol{x}^0 - \boldsymbol{x}^*\| + \frac{8\sqrt{\kappa_{\widehat{s}}}}{(\sqrt{\rho_{\widehat{s}}} + \sqrt{\ell_{\widehat{s}}})^2}\|\nabla_{\widehat{\mathcal{I}}} f(\boldsymbol{x}^*)\|,$$

*where $\widehat{\mathcal{I}} = \mathsf{supp}(\Phi_{3k}(\nabla f(\boldsymbol{x}^*))) \cup \mathsf{supp}(\boldsymbol{x}^*)$ and $T$ is the number of iterations.*

From this result, we can observe that for both sparsity- and rank-constrained quadratic loss minimization problems, AHSG-HT has a faster convergence rate $(1 - \frac{1}{18\sqrt{\kappa_{\widehat{s}}}})$ than the rate $(1 - \frac{1}{480\kappa_s})^{\frac{1}{2}}$ of HSG-HT. This is because the restricted condition number $\kappa_{\widehat{s}}$ is usually comparable to or even smaller than $\kappa_s$ since the factor $k$ in $\widehat{s} = 3k + k^*$ is allowed to be smaller than that in $s = 2k + k^*$ (explained below). Also, such an acceleration relaxes the restriction on the sparsity/low-rankness level $k$: AHSG-HT allows $k = \Omega(\kappa_{\widehat{s}}k^*)$ which is considerably superior to the condition of $k = \Omega(\kappa_s^2 k^*)$ as required in the analysis of other hard thresholding algorithms including HSG-HT, FG-HT and SVRG-HT. As a direct consequence, the statistical error bound $\mathcal{O}(\|\nabla_{\widehat{\mathcal{I}}} f(\boldsymbol{x}^*)\|)$ of AHSG-HT can be improved in the sense that the cardinality $|\widehat{\mathcal{I}}| = 3k + k^*$ has better dependency on the restricted condition number $\kappa_s$.

To better illustrate the boosted efficiency, we establish the computational complexity of AHSG-HT in IFO and hard thresholding in Corollary 2, whose proof is given in Appendix C.2.

**Corollary 2.** *Suppose the conditions in Theorem 3 hold. To achieve $\mathbb{E}\|\boldsymbol{x}^T - \boldsymbol{x}^*\| \leq \sqrt{\epsilon} + \frac{8\sqrt{\kappa_{\widehat{s}}}\|\nabla_{\widehat{\mathcal{I}}} f(\boldsymbol{x}^*)\|}{(\sqrt{\rho_{\widehat{s}}} + \sqrt{\ell_{\widehat{s}}})^2}$, the IFO complexity of AHSG-HT in Algorithm 1 is $\mathcal{O}\left(\frac{\sqrt{\kappa_{\widehat{s}}}}{\epsilon}\right)$ and the hard thresholding complexity is $\mathcal{O}\left(\sqrt{\kappa_{\widehat{s}}} \log\left(\frac{1}{\epsilon}\right)\right)$.*

Corollary 2 shows that equipped with heavy ball acceleration, the IFO complexity of HSG-HT in the quadratic case can be reduced from $\mathcal{O}\left(\frac{\kappa_s}{\epsilon}\right)$ to $\mathcal{O}\left(\frac{\sqrt{\kappa_{\widehat{s}}}}{\epsilon}\right)$, and its hard thresholding complexity can be reduced from $\mathcal{O}\left(\kappa_s \log\left(\frac{1}{\epsilon}\right)\right)$ to $\mathcal{O}\left(\sqrt{\kappa_{\widehat{s}}} \log\left(\frac{1}{\epsilon}\right)\right)$. Such an improvement in the dependency on restricted condition number is noteworthy in large-scale ill-conditioned learning problems.

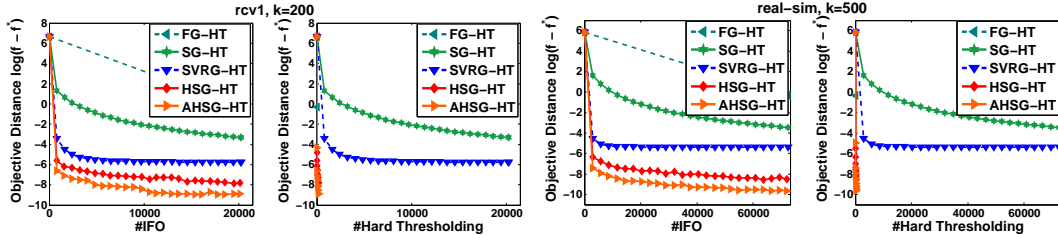

Figure 1: Single-epoch processing: comparison among hard thresholding algorithms for a single pass over data on sparse logistic regression with regularization parameter $\lambda = 10^{-5}$.

## 5 Experiments

We now compare the numerical performance of HSG-HT and AHSG-HT to several state-of-the-art algorithms, including FG-HT [13], SG-HT [20] and SVRG-HT [21, 22]. We evaluate all the considered algorithms on two sets of learning tasks. The first set contains two sparsity-constrained problems: logistic regression with $f_i(\boldsymbol{x}) = \log(1 + \exp(-\boldsymbol{b}_i \boldsymbol{a}_i^\top \boldsymbol{x})) + \frac{\lambda}{2}\|\boldsymbol{x}\|_2^2$ and multi-class softmax regression with $f_i(\boldsymbol{x}) = \sum_{j=1}^c \left[\frac{\lambda}{2c}\|\boldsymbol{x}_j\|_2^2 - \mathbf{1}\{\boldsymbol{b}_i = j\} \log \frac{\exp(\boldsymbol{a}_i^\top \boldsymbol{x}_j)}{\sum_{l=1}^c \exp(\boldsymbol{a}_i^\top \boldsymbol{x}_l)}\right]$, where $\boldsymbol{b}_i$ is the target output of $\boldsymbol{a}_i$ and $c$ is the class number. The second one is a rank-constrained linear regression problem:

$$\min_{\boldsymbol{x}} \frac{1}{n} \sum_{i=1}^n \left[\|\boldsymbol{b}_i - \langle \boldsymbol{x}, \boldsymbol{a}_i \rangle\|_2^2 + \frac{\lambda}{2}\|\boldsymbol{x}\|_F^2\right], \quad \text{s.t. } \mathsf{rank}\,(\boldsymbol{x}) \leq k,$$

which has several important applications including multi-class classification and multi-task regression for simultaneously learning shared characteristics of all classes/tasks [37], as well as high dimensional image and financial data modeling [5, 8]. We run simulations on six datasets, including rcv1, real-sim, mnist, news20, coil100 and caltech256. The details of these data sets are described in Appendix E. For HSG-HT and AHSG-HT, we follow our theory to exponentially expand the mini-batch size $s_t$ but with small exponential rate, with $\tau = 1$. Since there is no ground truth on real data, we run FG-HT sufficiently long until $\|x^t - x^{t+1}\|/\|x^t\| \leq 10^{-6}$, and then use the output $f(x^t)$ as the approximate optimal value $\hat{f}^*$ for sub-optimality estimation in Figure 1 and Figure 2.

**Single-epoch evaluation results.** We first consider the sparse logistic regression problem with single-epoch processing. As demonstrated in Figure 1 (more experiments in Appendix E) that HSG-HT and AHSG-HT converge significantly faster than the other considered algorithms in one pass over data. This confirms our theoretical predictions in Corollary 1 and 2 that HSG-HT and AHSG-HT are cheaper in IFO complexity than the sample-size-dependent algorithms when the desired accuracy is moderately small and data scale is large. In view of the hard thresholding complexity, AHSG-HT and HSG-HT are comparable to FG-HT and they all require much fewer hard thresholding operations than SG-HT and SVRG-HT to reach the same accuracy. This also well aligns with our theory: in one pass setting, roughly speaking, AHSG-HT and HSG-HT respectively need $\mathcal{O}\big(\sqrt{\kappa_{\hat{s}}}\log\big(\frac{n}{\kappa_{\hat{s}}}\big)\big)$ and $\mathcal{O}\big(\kappa_s\log\big(\frac{n}{\kappa_s}\big)\big)$ steps of hard thresholding which are both much less than the $\mathcal{O}(n)$ complexity of SG-HT and SVRG-HT. From Figure 1 and the magnifying figures in Appendix E for better displaying objective loss decrease along with hard thresholding iteration, one can observe that AHSG-HT has shaper convergence behavior than HSG-HT, which demonstrates the acceleration power of AHSG-HT.

**Multi-epoch evaluation results.** We further evaluate the considered algorithms on sparsity-constrained softmax regression and rank-constrained linear regression problems, for which an approach usually needs multiple cycles of data processing to reach high accuracy solution. In our implementation, HSG-HT (and AHSG-HT) degenerates to plain (and accelerated) FG-HT when the mini-batch size exceeds data size. The degeneration case, however, does not happen in our experiment with the specified small expanding rate. The corresponding results are illustrated in Figure 2, from which we can observe that HSG-HT and AHSG-HT exhibit much shaper convergence curves and lower hard thresholding complexity than other considered hard thresholding algorithms.

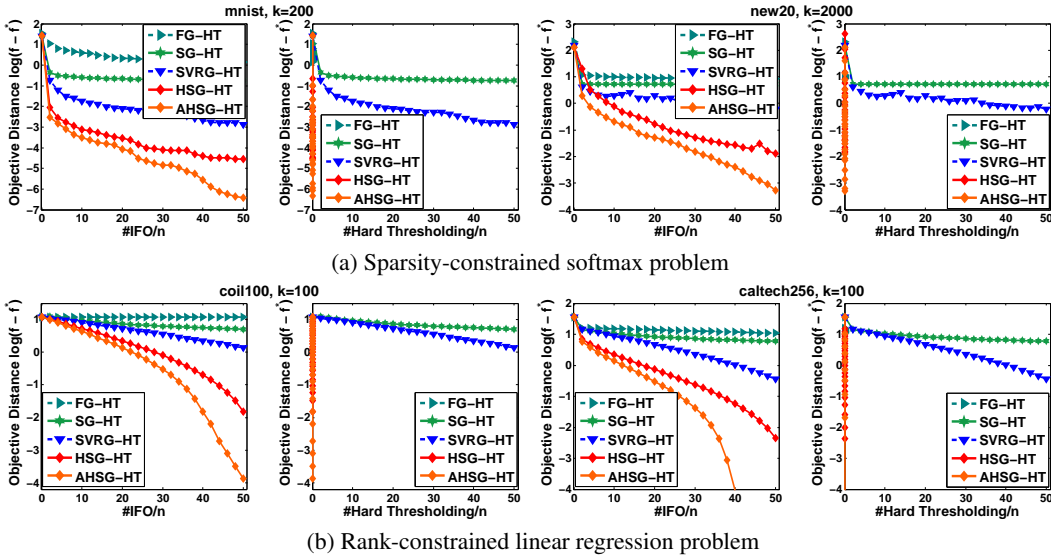

(a) Sparsity-constrained softmax problem

(b) Rank-constrained linear regression problem

Figure 2: Multi-epochs processing: comparison among hard thresholding algorithms with multiple passes over data for sparse softmax regression and rank-constrained linear regression problems, both with regularization parameters $\lambda = 10^{-5}$.

## 6 Conclusions

In this paper, we proposed HSG-HT as a hybrid stochastic gradient hard thresholding method for sparsity/rank-constrained empirical risk minimization problems. We proved that HSG-HT enjoys the

$\mathcal{O}\big(\kappa_s \log\big(\frac{1}{\epsilon}\big)\big)$ hard thresholding complexity like full gradient methods, while possessing sample-size-independent IFO complexity of $\mathcal{O}\big(\frac{\kappa_s}{\epsilon}\big)$. Compared to the existing variance-reduced hard thresholding algorithms, HSG-HT enjoys lower overall computational cost when sample size is large and the accuracy is moderately small. Furthermore, we showed that HSG-HT can be effectively accelerated via applying the heavy ball acceleration technique to attain improved dependency on restricted condition number. The provable efficiency of HSGT-HT and its accelerated variant has been confirmed by extensive numerical evaluation with comparison against the prior state-of-the-art algorithms.

## Acknowledgements

Jiashi Feng was partially supported by NUS startup R-263-000-C08-133, MOE Tier-I R-263-000-C21-112, NUS IDS R-263-000-C67-646, ECRA R-263-000-C87-133 and MOE Tier-II R-263-000-D17-112. Xiao-Tong Yuan was supported in part by Natural Science Foundation of China (NSFC) under Grant 61522308 and Grant 61876090, and in part by Tencent AI Lab Rhino-Bird Joint Research Program No.JR201801.

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
