[Supplementary Material]

# Efficient Stochastic Gradient Hard Thresholding

**Pan Zhou**[*]     **Xiao-Tong Yuan**[†]     **Jiashi Feng**[*]
[*] Learning & Vision Lab, National University of Singapore, Singapore
[†] B-DAT Lab, Nanjing University of Information Science & Technology, Nanjing, China
pzhou@u.nus.edu     xtyuan@nuist.edu.cn     elefjia@nus.edu.sg

## Abstract

This supplementary document contains the technical proofs of convergence results and some additional numerical results of the NeurIPS'18 paper entitled "Efficient Stochastic Gradient Hard Thresholding". It is structured as follows. The proof of the results in Section 3, including Theorems 1 and 2 and Corollary 1, is presented in Appendix B. Then Appendix C provides the proof of the results in Section 4, including Theorems 3 and Corollary 2. Next, Appendix D gives the proof of auxiliary lemmas. Finally, the detailed descriptions of datasets and more experimental results are provided in Appendix E.

## A   Auxiliary Lemmas

In this section, we introduce auxiliary lemmas which will be used for proving the results in the manuscript. For readability, we defer the proofs of some lemmas into Appendix D.

**Lemma 1.** *[1] When $\Phi_k(\boldsymbol{x}) : \mathbb{R}^d \to \mathbb{R}^d$ is a vector hard thresholding operator that keeps the largest (in absolute value) $k$ entries in $\boldsymbol{x}$ and sets other entries to zero, we have that for any $\boldsymbol{x}^*$ with $\|\boldsymbol{x}^*\|_0 = k^* < k$,*

$$\|\Phi_k(\boldsymbol{x}) - \boldsymbol{x}^*\|^2 \leq \left(1 + \frac{2\sqrt{k^*}}{\sqrt{k - k^*}}\right) \|\boldsymbol{x} - \boldsymbol{x}^*\|^2. \tag{3}$$

*On the other hand, when $\Phi_k(\boldsymbol{x}) : \mathbb{R}^{d_1 \times d_2} \to \mathbb{R}^{d_1 \times d_2}$ is a matrix hard thresholding operator that keeps the largest $k$ top singular values of matrix $\boldsymbol{x}$ and sets other singular values to zero, we have that the property (3) still holds for any $\boldsymbol{x}^*$ with $\mathsf{rank}(\boldsymbol{x}^*) = k^* < k$.*

**Lemma 2.** *[2] Let $\Phi_k(\boldsymbol{x}) : \mathbb{R}^d \to \mathbb{R}^d$ be a vector hard thresholding operator and $\boldsymbol{y} = \Phi_k(\boldsymbol{x})$. Then for any $\boldsymbol{y}^*$ with $\|\boldsymbol{y}^*\|_0 = k^* < k$, we have*

$$\|\boldsymbol{y} - \boldsymbol{x}\|^2 \leq \frac{d - k}{d - k^*} \|\boldsymbol{y}^* - \boldsymbol{x}\|^2.$$

*On the other hand, when $\Phi_k(\boldsymbol{x}) : \mathbb{R}^{d_1 \times d_2} \to \mathbb{R}^{d_1 \times d_2}$ is a matrix hard thresholding operator, we have that for any $\boldsymbol{y}^*$ with $\mathsf{rank}(\boldsymbol{y}^*) = k^* < k$,*

$$\|\boldsymbol{y} - \boldsymbol{x}\|^2 \leq \frac{r - k}{r - k^*} \|\boldsymbol{y}^* - \boldsymbol{x}\|^2.$$

*where $r = \mathsf{rank}(\boldsymbol{x})$.*

**Lemma 3.** *[3] Assume that $\boldsymbol{g}^t$ is the sampled gradient in Algorithm 1 for sparsity- or rank-constrained problem. Then the gradient variance of the gradient estimation $\boldsymbol{g}^t$ can be bounded as follows*

$$\mathbb{E}\|\boldsymbol{g}^t - \nabla f(\boldsymbol{x}^t)\|^2 \leq \frac{n - s_t}{n} \frac{1}{s_t} B_t,$$

*where $B_t = \frac{1}{n-1} \sum_{i=1}^n \|\nabla f_i(\boldsymbol{x}^t) - \nabla f(\boldsymbol{x}^t)\|^2$ and $\boldsymbol{x}^t$ denotes the variable at the $t$-th iteration.*

**Lemma 4.** *Assume that $\boldsymbol{g}^t$ is the sampled gradient in Algorithm 1 for the sparsity- or rank-constrained problem. Then we can bound $\mathbb{E}\|\boldsymbol{g}^t_{\mathcal{I}}\|^2$ as follows*

$$\mathbb{E}\|\boldsymbol{g}^t_{\mathcal{I}}\|^2 \leq \frac{3}{s_t}B_t + 6\ell_s\left(f(\boldsymbol{x}^t) - f(\boldsymbol{x}^*) + \langle\nabla_{\mathcal{I}}f(\boldsymbol{x}^*), \boldsymbol{x}^t - \boldsymbol{x}^*\rangle\right) + 3\|\nabla_{\mathcal{I}}f(\boldsymbol{x}^*)\|^2,$$

*where $B_t = \frac{1}{n-1}\sum_{i=1}^n \|\nabla f_i(\boldsymbol{x}^t) - \nabla f(\boldsymbol{x}^t)\|^2$, $\mathcal{I} = \mathsf{supp}(\boldsymbol{x}^*)\bigcup\mathsf{supp}(\boldsymbol{x}^t)\bigcup\mathsf{supp}(\boldsymbol{x}^{t+1})$ and $\ell_s$ denotes the smooth parameter with $s = 2k + k^*$. Here $k^*$ denotes the cardinality of the support set $\mathsf{supp}(\boldsymbol{x}^*)$.*

*Proof.* We defer the proof of Lemma 4 to Appendix D.1. $\qquad\square$

**Lemma 5.** *For both vector variable $\boldsymbol{x}$ in sparsity-constrained problem or matrix variable $\boldsymbol{x}$ in rank-constrained problem, we have*

$$\|\boldsymbol{x}^t - \boldsymbol{x}^*\|^2 \leq \frac{4}{\rho_s}(f(\boldsymbol{x}^*) - f(\boldsymbol{x}^t)) + \frac{8}{\rho_s^2}\|\boldsymbol{g}^t_{\mathcal{I}^t\cup\mathcal{I}^*}\|^2 + \frac{8}{\rho_s^2}\|\nabla f(\boldsymbol{x}^t) - \boldsymbol{g}^t\|^2,$$

*where $\mathcal{I}^t = \mathsf{supp}(\boldsymbol{x}^t)$ and $\mathcal{I}^* = \mathsf{supp}(\boldsymbol{x}^*)$.*

*Proof.* See the proof of Lemma 5 in Appendix D.2. $\qquad\square$

Let us denote $\rho(\boldsymbol{A})$ the spectral radius of a square matrix $\boldsymbol{A}$, i.e., the largest (in magnitude) eigenvalue of $\boldsymbol{A}$.

**Lemma 6.** *Let $\boldsymbol{A} \in \mathbb{R}^{d\times d}$. Assume that $\mu\boldsymbol{I} \preceq \boldsymbol{A} \preceq \ell\boldsymbol{I}$ for some $0 < \mu < \ell$. Then the following inequality holds*

$$\rho\left(\begin{bmatrix} (1+\nu)\boldsymbol{I} - \eta\boldsymbol{A} & -\nu\boldsymbol{I} \\ \boldsymbol{I} & 0 \end{bmatrix}\right) \leq \max\left\{|1 - \sqrt{\eta\mu}|, |1 - \sqrt{\eta\ell}|\right\}$$

*for $\nu = \max\left\{|1 - \sqrt{\eta\mu}|^2, |1 - \sqrt{\eta\ell}|^2\right\}$.*

*Proof.* We defer the proof of Lemma 6 to Appendix D.3. $\qquad\square$

# B  Proofs for Section 3

## B.1  Proof of Theorem 1

*Proof.* Here we also first consider the sparsity-constrained problem. Assume $\boldsymbol{v} = \boldsymbol{x}^t - \eta\boldsymbol{g}^t_{\mathcal{I}}$ and $\mathcal{I} = \mathcal{I}^* \cup \mathcal{I}^t \cup \mathcal{I}^{t+1}$, where $\mathcal{I}^* = \mathsf{supp}(\boldsymbol{x}^*)$, $\mathcal{I}^t = \mathsf{supp}(\boldsymbol{x}^t)$ and $\mathcal{I}^{t+1} = \mathsf{supp}(\boldsymbol{x}^{t+1})$. Then we have

$$
\begin{aligned}
&\mathbb{E}\|\boldsymbol{v} - \boldsymbol{x}^*\|^2 \\
=&\mathbb{E}\|\boldsymbol{x}^t - \eta\boldsymbol{g}^t_{\mathcal{I}} - \boldsymbol{x}^*\|^2 \\
=&\mathbb{E}\|\boldsymbol{x}^t - \boldsymbol{x}^*\|^2 + \eta^2\mathbb{E}\|\boldsymbol{g}^t_{\mathcal{I}}\|^2 - 2\eta\mathbb{E}\langle\boldsymbol{x}^t - \boldsymbol{x}^*, \boldsymbol{g}^t_{\mathcal{I}}\rangle \qquad\qquad (4) \\
\overset{①}{\leq}&\mathbb{E}\|\boldsymbol{x}^t - \boldsymbol{x}^*\|^2 + \eta^2\mathbb{E}\|\boldsymbol{g}^t_{\mathcal{I}}\|^2 - 2\eta\mathbb{E}\left(f(\boldsymbol{x}^t) - f(\boldsymbol{x}^*)\right) \\
\overset{②}{\leq}&\mathbb{E}\|\boldsymbol{x}^t - \boldsymbol{x}^*\|^2 + \eta^2\left[\frac{3}{s_t}B_t + 6\ell_s\left(f(\boldsymbol{x}^t) - f(\boldsymbol{x}^*) + \langle\nabla_{\mathcal{I}}f(\boldsymbol{x}^*), \boldsymbol{x}^t - \boldsymbol{x}^*\rangle\right) + 3\|\nabla_{\mathcal{I}}f(\boldsymbol{x}^*)\|^2\right] \\
&- 2\eta\mathbb{E}\left[f(\boldsymbol{x}^t) - f(\boldsymbol{x}^*)\right] \\
\overset{③}{=}&\mathbb{E}\|\boldsymbol{x}^t - \boldsymbol{x}^*\|^2 + 2\eta(3\eta\ell_s - 1)\left[f(\boldsymbol{x}^t) - f(\boldsymbol{x}^*)\right] + 6\eta^2\ell_s\langle\nabla_{\mathcal{I}}f(\boldsymbol{x}^*), \boldsymbol{x}^t - \boldsymbol{x}^*\rangle + \frac{3\eta^2}{s_t}B_t + 3\eta^2\|\nabla_{\mathcal{I}}f(\boldsymbol{x}^*)\|^2,
\end{aligned}
$$

where ① holds by using $\|(\boldsymbol{x}^t - \boldsymbol{x}^*)_{\mathcal{I}^c}\| = 0$ and the convexity of $f(\boldsymbol{x})$, namely, $f(\boldsymbol{x}^*) - f(\boldsymbol{x}^t) \geq \langle\boldsymbol{x}^* - \boldsymbol{x}^t, \nabla f(\boldsymbol{x}^t)\rangle = \langle\boldsymbol{x}^* - \boldsymbol{x}^t, \nabla_{\mathcal{I}}f(\boldsymbol{x}^t)\rangle$, and ② holds by using Lemma 4, and ③ holds by using $\|\nabla_{\mathcal{I}}f(\boldsymbol{x}^*)\|_2^2 \leq \|\nabla_{\widetilde{\mathcal{I}}}f(\boldsymbol{x}^*)\|_2^2$ where $\widetilde{\mathcal{I}} = \mathsf{supp}(\Phi_{2k}(\nabla f(\boldsymbol{x}^*))) \cup \mathsf{supp}(\boldsymbol{x}^*)$.

Next, we apply Lemma 1 and obtain

$$\mathbb{E}\|\boldsymbol{x}^{t+1} - \boldsymbol{x}^*\|^2 \leq \alpha \mathbb{E}\|\boldsymbol{v} - \boldsymbol{x}^*\|^2 \tag{5}$$

$$\leq \alpha \left[ \mathbb{E}\|\boldsymbol{x}^t - \boldsymbol{x}^*\|^2 + 2\eta(3\eta\ell - 1)\left[ f(\boldsymbol{x}^t) - f(\boldsymbol{x}^*)\right] + 6\eta^2\ell_s \langle \nabla_{\mathcal{I}} f(\boldsymbol{x}^*), \boldsymbol{x}^t - \boldsymbol{x}^*\rangle \right]$$

$$+ \alpha \left[ \frac{3\eta^2}{s_t} B_t + 3\eta^2 \|\nabla_{\mathcal{I}} f(\boldsymbol{x}^*)\|^2 \right],$$

where $\alpha = 1 + \frac{2\sqrt{k^*}}{\sqrt{k-k^*}}$. On the other hand, we have

$$f(\boldsymbol{x}^t) - f(\boldsymbol{x}^*) \geq \langle \nabla f(\boldsymbol{x}^*), \boldsymbol{x}^t - \boldsymbol{x}^*\rangle + \frac{\rho_s}{2}\|\boldsymbol{x}^t - \boldsymbol{x}^*\|^2. \tag{6}$$

By setting $\eta \leq \frac{1}{3\ell_s}$ and plugging Eqn. (6) into Eqn. (5), we can obtain

$$\mathbb{E}\|\boldsymbol{x}^{t+1} - \boldsymbol{x}^*\|^2$$

$$\leq \alpha\left[1 + \rho\eta(3\eta\ell_s - 1)\right] \mathbb{E}\|\boldsymbol{x}^t - \boldsymbol{x}^*\|^2 + 2\alpha\eta(6\eta\ell_s - 1)\langle \nabla f(\boldsymbol{x}^*), \boldsymbol{x}^t - \boldsymbol{x}^*\rangle + \frac{3\alpha\eta^2}{s_t} B_t + 3\alpha\eta^2 \|\nabla_{\mathcal{I}} f(\boldsymbol{x}^*)\|^2$$

$$\overset{①}{\leq} \alpha\left[1 + \rho\eta(3\eta\ell_s - 1)\right] \mathbb{E}\|\boldsymbol{x}^t - \boldsymbol{x}^*\|^2 + \frac{3\alpha\eta^2}{s_t} B + 3\alpha\eta^2 \|\nabla_{\mathcal{I}} f(\boldsymbol{x}^*)\|^2,$$

$$\tag{7}$$

where ① holds since we set $\eta = \frac{1}{6\ell_s}$ and $B = \max_t B_t$. Then we let

$$\beta := \alpha\left[1 + \rho\eta(3\eta\ell_s - 1)\right] = \alpha\left(1 - \frac{1}{12\kappa_s}\right) \in (0,1) \quad \text{where} \quad \kappa_s = \frac{\ell_s}{\rho_s}.$$

We further set $s_t = \tau/\omega^t$ and assume that $\tau$ is large enough such that

$$\gamma := \frac{3\alpha\eta^2 B}{\tau} = \frac{\alpha B}{12\tau\ell_s^2} \leq \delta\|\boldsymbol{x}^0 - \boldsymbol{x}^*\|^2, \tag{8}$$

where $\delta$ is a positive constant and will be discussed later. Now we use mathematical induction to prove

$$\mathbb{E}\|\boldsymbol{x}^t - \boldsymbol{x}^*\|^2 \leq \theta^t \|\boldsymbol{x}^0 - \boldsymbol{x}^*\|^2 + \frac{\alpha}{12(1-\beta)\ell_s^2} \|\nabla_{\mathcal{I}} f(\boldsymbol{x}^*)\|^2, \ (\forall t), \tag{9}$$

where $\theta < 1$ is a constant and will be given below.

Obviously, when $t = 0$, Eqn. (9) holds. Now assume that for all $k \leq t$, Eqn. (9) holds. Then for $k = t + 1$, we have

$$\mathbb{E}\|\boldsymbol{x}^{t+1} - \boldsymbol{x}^*\|^2 \leq \beta \mathbb{E}\|\boldsymbol{x}^t - \boldsymbol{x}^*\|^2 + \delta\omega^t \|\boldsymbol{x}^0 - \boldsymbol{x}^*\|^2 + \frac{\alpha}{12\ell_s^2} \|\nabla_{\mathcal{I}} f(\boldsymbol{x}^*)\|^2$$

$$\leq (\beta\theta^t + \delta\omega^t)\|\boldsymbol{x}^0 - \boldsymbol{x}^*\|^2 + \frac{\alpha}{12\ell_s^2}\left[\frac{\beta}{1-\beta} + 1\right] \|\nabla_{\mathcal{I}} f(\boldsymbol{x}^*)\|_2^2$$

$$\overset{①}{\leq} \theta^{t+1}\|\boldsymbol{x}^0 - \boldsymbol{x}^*\|^2 + \frac{\alpha}{12(1-\beta)\ell_s^2} \|\nabla_{\mathcal{I}} f(\boldsymbol{x}^*)\|_2^2$$

where ① holds since we let

$$\omega = \theta = \beta + \delta. \tag{10}$$

This means that if Eqn. (8) holds, then Eqn. (9) always holds. So the conclusion holds.

Finally, we discuss the value of $k$ such that $\beta \in (0,1)$. It is easily to check that if $k \geq \left(\frac{6400}{9}\kappa_s^2 + 1\right)k^*$, where $\kappa_s = \ell_s/\rho_s$, then we have $\beta = \alpha\left(1 - \frac{1}{12\kappa_s}\right) < 1 - \frac{1}{120\kappa_s} - \frac{1}{160\kappa_s^2}$. So we just let

$$k \geq \left(712\kappa_s^2 + 1\right)k^*, \quad \text{and} \quad \beta = \alpha\left(1 - \frac{1}{12\kappa_s}\right) \leq 1 - \frac{1}{120\kappa_s}.$$

Then let $\tau \geq \frac{40\alpha B}{3\ell_s \rho_s \|\boldsymbol{x}^0 - \boldsymbol{x}^*\|^2}$. We have

$$\omega = \theta = \beta + \delta \leq 1 - \frac{1}{120\kappa_s} + \frac{3}{480\kappa_s} = 1 - \frac{1}{480\kappa_s}.$$

Therefore, we have

$$\mathbb{E}\|\boldsymbol{x}^t - \boldsymbol{x}^*\|^2 \leq \theta^t \|\boldsymbol{x}^0 - \boldsymbol{x}^*\|^2 + \frac{\alpha}{12(1-\beta)\ell_s^2}\|\nabla_{\mathcal{I}} f(\boldsymbol{x}^*)\|^2, \ (\forall t), \tag{11}$$

where $\beta = \alpha\left(1 - \frac{1}{12\kappa_s}\right) \leq 1 - \frac{1}{120\kappa_s}$ and $\theta \leq 1 - \frac{1}{480\kappa_s}$. Then we can further derive

$$\mathbb{E}\|\boldsymbol{x}^T - \boldsymbol{x}^*\| \leq \sqrt{\mathbb{E}\|\boldsymbol{x}^T - \boldsymbol{x}^*\|^2} \leq \sqrt{\theta^T\|\boldsymbol{x}^0 - \boldsymbol{x}^*\|^2 + \frac{\alpha}{12(1-\beta)\ell_s^2}\|\nabla_{\widetilde{\mathcal{I}}} f(\boldsymbol{x}^*)\|^2}$$

$$\leq \theta^{\frac{T}{2}}\|\boldsymbol{x}^0 - \boldsymbol{x}^*\| + \frac{\sqrt{\alpha}}{\ell_s\sqrt{12(1-\beta)}}\|\nabla_{\widetilde{\mathcal{I}}} f(\boldsymbol{x}^*)\|,$$

where $\theta \leq 1 - \frac{1}{480\kappa_s}$. So we obtain the result on sparsity-constrained problem.

For rank-constrained problem, its proof is very similar to the proof for sparsity-constrained problem. Firstly, assume that the skinny SVDs of $\boldsymbol{x}^t$, $\boldsymbol{x}^{t+1}$, $\boldsymbol{x}^*$ are respectively $\boldsymbol{x}^t = \boldsymbol{U}_t \boldsymbol{\Sigma}_t \boldsymbol{V}_t^T$, $\boldsymbol{x}^{t+1} = \boldsymbol{U}_{t+1}\boldsymbol{\Sigma}_{t+1}\boldsymbol{V}_{t+1}^T$ and $\boldsymbol{x}^* = \boldsymbol{U}_*\boldsymbol{\Sigma}_*\boldsymbol{V}_*^T$. Then we have $\boldsymbol{x}^t = \boldsymbol{U}_t\boldsymbol{U}_t^T\boldsymbol{x}^t$, $\boldsymbol{x}^{t+1} = \boldsymbol{U}_{t+1}\boldsymbol{U}_{t+1}^T\boldsymbol{x}^{t+1}$ and $\boldsymbol{x}^* = \boldsymbol{U}_*\boldsymbol{U}_*^T\boldsymbol{x}^*$. We further define the projection operation $\mathcal{P}_{\boldsymbol{U}}(\boldsymbol{x}) = \boldsymbol{U}\boldsymbol{U}^T\boldsymbol{x}$ which projects $\boldsymbol{x}$ in the subspace spanned by $\boldsymbol{U}$. Then let $\boldsymbol{v} = \boldsymbol{x}^t - \eta\boldsymbol{g}_{\mathcal{I}}^t$ and $\mathcal{I} = \mathcal{I}^* \cup \mathcal{I}^t \cup \mathcal{I}^{t+1}$, where $\mathcal{I}^* = \mathsf{supp}(\boldsymbol{x}^*)$, $\mathcal{I}^t = \mathsf{supp}(\boldsymbol{x}^t)$ and $\mathcal{I}^{t+1} = \mathsf{supp}(\boldsymbol{x}^{t+1})$. The notation $\boldsymbol{x}_{\mathcal{I}^t}^t = \boldsymbol{U}_t\boldsymbol{U}_t^T\boldsymbol{x}^t = \mathcal{P}_{\boldsymbol{U}^t}(\boldsymbol{x}^t)$. For the specifical notation $\boldsymbol{g}_{\mathcal{I}}^t$, it denotes $\boldsymbol{g}_{\mathcal{I}}^t = \mathcal{P}_{\mathcal{I}}(\boldsymbol{g}^t) = (\boldsymbol{U}_t\boldsymbol{U}_t^T + \boldsymbol{U}_{t+1}\boldsymbol{U}_{t+1}^T + \boldsymbol{U}_*\boldsymbol{U}_*^T - \boldsymbol{U}_t\boldsymbol{U}_t^T\boldsymbol{U}_{t+1}\boldsymbol{U}_{t+1}^T - \boldsymbol{U}_t\boldsymbol{U}_t^T\boldsymbol{U}_*\boldsymbol{U}_*^T - \boldsymbol{U}_{t+1}\boldsymbol{U}_{t+1}^T\boldsymbol{U}_*\boldsymbol{U}_*^T + \boldsymbol{U}_t\boldsymbol{U}_t^T\boldsymbol{U}_{t+1}\boldsymbol{U}_{t+1}^T\boldsymbol{U}_*\boldsymbol{U}_*^T)\boldsymbol{g}^t$ where $\mathcal{P}_{\mathcal{I}}$ is a projection.

Then we prove the result similar to Eqn. (4) as follows:

$$\mathbb{E}\|\boldsymbol{v} - \boldsymbol{x}^*\|^2$$
$$=\mathbb{E}\|\boldsymbol{x}^t - \eta\boldsymbol{g}_{\mathcal{I}}^t - \boldsymbol{x}^*\|^2$$
$$=\mathbb{E}\|\boldsymbol{x}^t - \boldsymbol{x}^*\|^2 + \eta^2\mathbb{E}\|\boldsymbol{g}_{\mathcal{I}}^t\|^2 - 2\eta\mathbb{E}\langle \boldsymbol{x}^t - \boldsymbol{x}^*, \boldsymbol{g}_{\mathcal{I}}^t\rangle$$
$$\overset{\text{①}}{\leq}\mathbb{E}\|\boldsymbol{x}^t - \boldsymbol{x}^*\|^2 + \eta^2\mathbb{E}\|\boldsymbol{g}_{\mathcal{I}}^t\|^2 - 2\eta\mathbb{E}\left(f(\boldsymbol{x}^t) - f(\boldsymbol{x}^*)\right)$$
$$\overset{\text{②}}{\leq}\mathbb{E}\|\boldsymbol{x}^t - \boldsymbol{x}^*\|^2 + \eta^2\left[\frac{3}{s_t}B_t + 6\ell_s\left(f(\boldsymbol{x}^t) - f(\boldsymbol{x}^*) + \langle\nabla_{\mathcal{I}} f(\boldsymbol{x}^*), \boldsymbol{x}^t - \boldsymbol{x}^*\rangle\right) + 3\|\nabla_{\mathcal{I}} f(\boldsymbol{x}^*)\|^2\right]$$
$$\quad - 2\eta\mathbb{E}\left[f(\boldsymbol{x}^t) - f(\boldsymbol{x}^*)\right]$$
$$\overset{\text{③}}{=}\mathbb{E}\|\boldsymbol{x}^t - \boldsymbol{x}^*\|^2 + 2\eta(3\eta\ell_s - 1)\left[f(\boldsymbol{x}^t) - f(\boldsymbol{x}^*)\right] + 6\eta^2\ell_s\langle\nabla_{\mathcal{I}} f(\boldsymbol{x}^*), \boldsymbol{x}^t - \boldsymbol{x}^*\rangle + \frac{3\eta^2}{s_t}B_t + 3\eta^2\|\nabla_{\mathcal{I}} f(\boldsymbol{x}^*)\|^2,$$

where ① holds since we have $\mathcal{P}_{\mathcal{I}}(\boldsymbol{x}^t - \boldsymbol{x}^*) = (\boldsymbol{U}_t\boldsymbol{U}_t^T + \boldsymbol{U}_{t+1}\boldsymbol{U}_{t+1}^T + \boldsymbol{U}_*\boldsymbol{U}_*^T - \boldsymbol{U}_t\boldsymbol{U}_t^T\boldsymbol{U}_{t+1}\boldsymbol{U}_{t+1}^T - \boldsymbol{U}_t\boldsymbol{U}_t^T\boldsymbol{U}_*\boldsymbol{U}_*^T - \boldsymbol{U}_{t+1}\boldsymbol{U}_{t+1}^T\boldsymbol{U}_*\boldsymbol{U}_*^T + \boldsymbol{U}_t\boldsymbol{U}_t^T\boldsymbol{U}_{t+1}\boldsymbol{U}_{t+1}^T\boldsymbol{U}_*\boldsymbol{U}_*^T)(\boldsymbol{U}_t\boldsymbol{U}_t^T\boldsymbol{x}^t - \boldsymbol{U}_*\boldsymbol{U}_*^T\boldsymbol{x}^*) = \boldsymbol{x}^t - \boldsymbol{x}^*$ which gives $\|(\boldsymbol{x}^t - \boldsymbol{x}^*)_{\mathcal{I}^c}\| = \|(\boldsymbol{I} - \mathcal{P}_{\mathcal{I}})(\boldsymbol{x}^t - \boldsymbol{x}^*)\| = 0$ and $\mathbb{E}\langle\boldsymbol{x}^t - \boldsymbol{x}^*, \boldsymbol{g}_{\mathcal{I}}^t\rangle = \mathbb{E}\langle\boldsymbol{x}^t - \boldsymbol{x}^*, \mathcal{P}_{\mathcal{I}}(\boldsymbol{g}^t)\rangle = \mathbb{E}\langle\mathcal{P}_{\mathcal{I}}(\boldsymbol{x}^t - \boldsymbol{x}^*), \mathcal{P}_{\mathcal{I}}(\boldsymbol{g}^t)\rangle = \mathbb{E}\langle\boldsymbol{x}^t - \boldsymbol{x}^*, \boldsymbol{g}^t\rangle = \mathbb{E}\langle\boldsymbol{x}^t - \boldsymbol{x}^*, \nabla f(\boldsymbol{x}^t)\rangle \leq f(\boldsymbol{x}^*) - f(\boldsymbol{x}^t)$. ② still uses the results in Lemma 4. ③ holds by using $\|\nabla_{\mathcal{I}} f(\boldsymbol{x}^*)\|_2^2 \leq \|\nabla_{\widetilde{\mathcal{I}}} f(\boldsymbol{x}^*)\|_2^2$ where $\widetilde{\mathcal{I}} = \mathsf{supp}(\Phi_{2k}(\nabla f(\boldsymbol{x}^*))) \cup \mathsf{supp}(\boldsymbol{x}^*)$.

Before we apply Lemma 1 to obtain similar result in Eqn. (5). We first establish

$$\Phi_k(\boldsymbol{v}) = \boldsymbol{U}_{t+1}\boldsymbol{U}_{t+1}^T(\boldsymbol{x}^t - \eta\boldsymbol{g}_{\mathcal{I}}^t) = \boldsymbol{U}_{t+1}\boldsymbol{U}_{t+1}^T\boldsymbol{x}^t - \eta\boldsymbol{U}_{t+1}\boldsymbol{U}_{t+1}^T[(\boldsymbol{U}_t\boldsymbol{U}_t^T + \boldsymbol{U}_{t+1}\boldsymbol{U}_{t+1}^T + \boldsymbol{U}_*\boldsymbol{U}_*^T$$
$$\quad - \boldsymbol{U}_t\boldsymbol{U}_t^T\boldsymbol{U}_{t+1}\boldsymbol{U}_{t+1}^T - \boldsymbol{U}_t\boldsymbol{U}_t^T\boldsymbol{U}_*\boldsymbol{U}_*^T - \boldsymbol{U}_{t+1}\boldsymbol{U}_{t+1}^T\boldsymbol{U}_*\boldsymbol{U}_*^T + \boldsymbol{U}_t\boldsymbol{U}_t^T\boldsymbol{U}_{t+1}\boldsymbol{U}_{t+1}^T\boldsymbol{U}_*\boldsymbol{U}_*^T)]\boldsymbol{g}^t$$
$$=\boldsymbol{U}_{t+1}\boldsymbol{U}_{t+1}^T\boldsymbol{x}^t - \eta\boldsymbol{U}_{t+1}\boldsymbol{U}_{t+1}^T\boldsymbol{g}^t = \boldsymbol{U}_{t+1}\boldsymbol{U}_{t+1}^T(\boldsymbol{x}^t - \eta\boldsymbol{g}^t) = \Phi_k(\boldsymbol{x}^t - \eta\boldsymbol{g}^t) = \boldsymbol{x}^{t+1}. \tag{12}$$

Then we apply Lemma 1 to establish

$$\|\boldsymbol{x}^{t+1} - \boldsymbol{x}^*\|^2 = \|\Phi_k(\boldsymbol{v}) - \boldsymbol{x}^*\|^2 \leq \left(1 + \frac{2\sqrt{k^*}}{\sqrt{k - k^*}}\right)\|\boldsymbol{v} - \boldsymbol{x}^*\|^2$$

$$\leq \alpha\left[\mathbb{E}\|\boldsymbol{x}^t - \boldsymbol{x}^*\|^2 + 2\eta(3\eta\ell - 1)\left[f(\boldsymbol{x}^t) - f(\boldsymbol{x}^*)\right] + 6\eta^2\ell_s\langle\nabla_{\mathcal{I}}f(\boldsymbol{x}^*), \boldsymbol{x}^t - \boldsymbol{x}^*\rangle\right]$$

$$+ \alpha\left[\frac{3\eta^2}{s_t}B_t + 3\eta^2\|\nabla_{\mathcal{I}}f(\boldsymbol{x}^*)\|^2\right],$$

where $\alpha = 1 + \frac{2\sqrt{k^*}}{\sqrt{k - k^*}}$.

Next by using Assumption 1 in which we have $f(\boldsymbol{x}^t) - f(\boldsymbol{x}^*) \geq \langle\nabla f(\boldsymbol{x}^*), \boldsymbol{x}^t - \boldsymbol{x}^*\rangle + \frac{\rho_s}{2}\|\boldsymbol{x}^t - \boldsymbol{x}^*\|^2$, we can establish the result in Eqn. (7). Finally, since the deduction after Eqn. (7) does not rely on the rank-constrained problem, we can just follow the above proof and obtain the desired result on rank-constrained problem. The proof is completed. $\square$

## B.2 Proof of Corollary 1

*Proof.* Here we use the result in Eqn. (11) in Section B.1:

$$\mathbb{E}\|\boldsymbol{x}^t - \boldsymbol{x}^*\|^2 \leq \theta^t\|\boldsymbol{x}^0 - \boldsymbol{x}^*\|^2 + \frac{\alpha}{12(1 - \beta)\ell_s^2}\|\nabla_{\mathcal{I}}f(\boldsymbol{x}^*)\|^2, \ (\forall t),$$

where $\theta \leq 1 - \frac{1}{480\kappa_s}$ and $\beta = \alpha\left(1 - \frac{1}{12\kappa_s}\right)$. Then to achieve $\theta^T\|\boldsymbol{x}^0 - \boldsymbol{x}^*\|^2 \leq \epsilon$, we have

$$T \geq \log_{1/\theta}\left(\frac{\|\boldsymbol{x}^0 - \boldsymbol{x}^*\|^2}{\epsilon}\right)$$

In this way, we have

$$\mathbb{E}\|\boldsymbol{x}^T - \boldsymbol{x}^*\|_2 \leq \sqrt{\mathbb{E}\|\boldsymbol{x}^T - \boldsymbol{x}^*\|_2^2} \leq \sqrt{\theta^T\|\boldsymbol{x}^0 - \boldsymbol{x}^*\|^2 + \frac{\alpha}{12(1 - \beta)\ell_s^2}\|\nabla_{\widetilde{\mathcal{I}}}f(\boldsymbol{x}^*)\|^2}$$

$$\leq \sqrt{\epsilon + \frac{\alpha}{12(1 - \beta)\ell_s^2}\|\nabla_{\widetilde{\mathcal{I}}}f(\boldsymbol{x}^*)\|_2^2} \leq \sqrt{\epsilon} + \frac{\sqrt{\alpha}}{\ell_s\sqrt{12(1 - \beta)}}\|\nabla_{\widetilde{\mathcal{I}}}f(\boldsymbol{x}^*)\|.$$

Therefore, the IFO complexity is

$$\tau\left[1 + \frac{1}{\omega} + \cdots + \frac{1}{\omega^{T-1}}\right] = \tau\frac{(1/\omega)^{\log_{1/\theta}\left(\frac{\|\boldsymbol{x}^0 - \boldsymbol{x}^*\|^2}{\epsilon}\right)} - 1}{1/\omega - 1} \overset{①}{=} \frac{\tau}{1/\omega - 1}\left[\frac{\|\boldsymbol{x}^0 - \boldsymbol{x}^*\|^2}{\epsilon} - 1\right]$$

$$\overset{②}{\leq} \frac{6400\alpha B}{\rho_s\ell_s} \cdot \frac{\kappa_s}{\epsilon} \overset{③}{=} \mathcal{O}\left(\frac{\kappa_s}{\epsilon}\right),$$

where ① uses $\omega = \theta = 1 - \frac{1}{480\kappa_s}$; ② uses $\tau \geq \frac{40\alpha B}{3\rho_s\ell_s\|\boldsymbol{x}^0 - \boldsymbol{x}^*\|^2}$; ③ holds since (1) the parameter $\rho_s$ is the strong convex parameter at sparsity/low-rank level $s$ and thus is not very small since $s$ is much smaller than the feature dimension, (2) we can always scale the problem such that $\rho_s$ is not small. Notice, such a scale does not affect the ratio $B/\ell_s$ since they always scale at the same order. Thus, we have the IFO complexity $\mathcal{O}\left(\frac{\kappa_s}{\epsilon}\right)$.

On the other hand, we have

$$\log_{1/\theta}\left(\frac{1}{\epsilon}\right) = \frac{\log\left(\frac{1}{\epsilon}\right)}{\log\left(\frac{1}{\theta}\right)} = \frac{\log\left(\frac{1}{\epsilon}\right)}{\log\left(1 + \frac{1}{480\kappa_s - 1}\right)} = \frac{\log\left(\frac{1}{\epsilon}\right)}{\log\left(1 + \frac{1}{480\kappa_s - 1}\right)} \overset{①}{\leq} \mathcal{O}\left(\kappa_s\log\left(\frac{1}{\epsilon}\right)\right),$$

where  holds since we have $\log(1 + x) \geq \log(2) \cdot x$ for $x \in [0, 1]$. The proof is completed. $\square$

## B.3 Proof of Theorem 2

*Proof.* We first prove the result for sparsity-constrained problem. In this case, the variable $\boldsymbol{x}$ is vector. Let $\mathcal{I} = \mathcal{I}^{t+1} \cup \mathcal{I}^t \cup \mathcal{I}^*$, where $\mathcal{I}^* = \mathsf{supp}(\boldsymbol{x}^*)$, $\mathcal{I}^t = \mathsf{supp}(\boldsymbol{x}^t)$ and $\mathcal{I}^{t+1} = \mathsf{supp}(\boldsymbol{x}^{t+1})$. Recall

that $\boldsymbol{x}^{t+1} = \Phi_k(\boldsymbol{x}^t - \eta \boldsymbol{g}^t)$. Then we have

$$f(\boldsymbol{x}^{t+1})$$

$$\leq f(\boldsymbol{x}^t) + \langle \nabla f(\boldsymbol{x}^t), \boldsymbol{x}^{t+1} - \boldsymbol{x}^t \rangle + \frac{\ell_s}{2}\|\boldsymbol{x}^{t+1} - \boldsymbol{x}^t\|^2$$

$$\leq f(\boldsymbol{x}^t) + \langle \boldsymbol{g}^t, \boldsymbol{x}^{t+1} - \boldsymbol{x}^t \rangle + \frac{\ell_s}{2}\|\boldsymbol{x}^{t+1} - \boldsymbol{x}^t\|^2 + \|\nabla f(\boldsymbol{x}^t) - \boldsymbol{g}^t\|\|\boldsymbol{x}^{t+1} - \boldsymbol{x}^t\|$$

$$= f(\boldsymbol{x}^t) + \frac{1}{2\eta}\|\boldsymbol{x}_{\mathcal{I}}^{t+1} - \boldsymbol{x}_{\mathcal{I}}^t + \eta \boldsymbol{g}_{\mathcal{I}}^t\|^2 - \frac{\eta\|\boldsymbol{g}_{\mathcal{I}}^t\|^2}{2} - \frac{1 - \eta\ell_s}{2\eta}\|\boldsymbol{x}^{t+1} - \boldsymbol{x}^t\|^2 + \|\nabla f(\boldsymbol{x}^t) - \boldsymbol{g}^t\|\|\boldsymbol{x}^{t+1} - \boldsymbol{x}^t\|$$

$$= f(\boldsymbol{x}^t) + \frac{1}{2\eta}\|\boldsymbol{x}_{\mathcal{I}}^{t+1} - \boldsymbol{x}_{\mathcal{I}}^t + \eta \boldsymbol{g}_{\mathcal{I}}^t\|^2 - \frac{\eta\|\boldsymbol{g}_{\mathcal{I}\setminus(\mathcal{I}^t \cup \mathcal{I}^*)}^t\|^2}{2} - \frac{\eta\|\boldsymbol{g}_{\mathcal{I}^t \cup \mathcal{I}^*}^t\|^2}{2} - \frac{1 - \eta\ell_s}{2\eta}\|\boldsymbol{x}^{t+1} - \boldsymbol{x}^t\|^2$$

$$\quad + \|\nabla f(\boldsymbol{x}^t) - \boldsymbol{g}^t\|\|\boldsymbol{x}^{t+1} - \boldsymbol{x}^t\|$$

$$= f(\boldsymbol{x}^t) + \frac{1}{2\eta}\left(\|\boldsymbol{x}_{\mathcal{I}}^{t+1} - \boldsymbol{x}_{\mathcal{I}}^t + \eta \boldsymbol{g}_{\mathcal{I}}^t\|^2 - \eta^2\|\boldsymbol{g}_{\mathcal{I}\setminus(\mathcal{I}^t \cup \mathcal{I}^*)}^t\|^2\right) - \frac{\eta\|\boldsymbol{g}_{\mathcal{I}^t \cup \mathcal{I}^*}^t\|^2}{2} - \frac{1 - \eta\ell_s}{2\eta}\|\boldsymbol{x}^{t+1} - \boldsymbol{x}^t\|^2$$

$$\quad + \|\nabla f(\boldsymbol{x}^t) - \boldsymbol{g}^t\|\|\boldsymbol{x}^{t+1} - \boldsymbol{x}^t\|. \tag{13}$$

Now we bound the second term $\|\boldsymbol{x}_{\mathcal{I}}^{t+1} - \boldsymbol{x}_{\mathcal{I}}^t + \eta \boldsymbol{g}_{\mathcal{I}}^t\|^2 - \eta^2\|\boldsymbol{g}_{\mathcal{I}\setminus(\mathcal{I}^t \cup \mathcal{I}^*)}^t\|^2$. Here we adopt similar strategy in [2]. Since we have $\mathcal{I} \setminus (\mathcal{I}^t \cup \mathcal{I}^*) = \mathcal{I}^{t+1} \setminus (\mathcal{I}^t \cup \mathcal{I}^*) \subseteq \mathcal{I}^{t+1}$, then we can establish $\boldsymbol{x}_{\mathcal{I}\setminus(\mathcal{I}^t \cup \mathcal{I}^*)}^{t+1} = \boldsymbol{x}_{\mathcal{I}\setminus(\mathcal{I}^t \cup \mathcal{I}^*)}^t - \eta \boldsymbol{g}_{\mathcal{I}\setminus(\mathcal{I}^t \cup \mathcal{I}^*)}^t$. On the other hand, we have $\boldsymbol{x}_{\mathcal{I}\setminus(\mathcal{I}^t \cup \mathcal{I}^*)}^t = 0$ which further yields $\boldsymbol{x}_{\mathcal{I}\setminus(\mathcal{I}^t \cup \mathcal{I}^*)}^{t+1} = -\eta \boldsymbol{g}_{\mathcal{I}\setminus(\mathcal{I}^t \cup \mathcal{I}^*)}^t$. Next, we choose a set $\mathcal{R} \subseteq \mathcal{I}^t \setminus \mathcal{I}^{t+1}$ such that $|\mathcal{R}| = |\mathcal{I}^{t+1} \setminus (\mathcal{I}^t \cup \mathcal{I}^*)|$. We can find such a set $\mathcal{R}$ because we have $|\mathcal{I}^{t+1} \setminus (\mathcal{I}^t \cup \mathcal{I}^*)| = |\mathcal{I}^t \setminus \mathcal{I}^{t+1}| - |(\mathcal{I}^{t+1} \cap \mathcal{I}^*) \setminus \mathcal{I}^t|$. Besides, since $\boldsymbol{x}^{t+1} = \Phi_k(\boldsymbol{x}^t - \eta \boldsymbol{g}^t)$, we can establish:

$$\eta^2\|\boldsymbol{g}_{\mathcal{I}\setminus(\mathcal{I}^t \cup \mathcal{I}^*)}^t\|^2 = \|\boldsymbol{x}_{\mathcal{I}\setminus(\mathcal{I}^t \cup \mathcal{I}^*)}^{t+1}\|^2 \geq \|\boldsymbol{x}_{\mathcal{R}}^t - \eta \boldsymbol{g}_{\mathcal{R}}^t\|^2. \tag{14}$$

Then combining Eqn. (14) and the fact that $\boldsymbol{x}_{\mathcal{R}}^{t+1} = 0$, we have

$$\|\boldsymbol{x}_{\mathcal{I}}^{t+1} - \boldsymbol{x}_{\mathcal{I}}^t + \eta \boldsymbol{g}_{\mathcal{I}}^t\|^2 - \eta^2\|\boldsymbol{g}_{\mathcal{I}\setminus(\mathcal{I}^t \cup \mathcal{I}^*)}^t\|^2 \leq \|\boldsymbol{x}_{\mathcal{I}}^{t+1} - \boldsymbol{x}_{\mathcal{I}}^t + \eta \boldsymbol{g}_{\mathcal{I}}^t\|^2 - \|\boldsymbol{x}_{\mathcal{R}}^t - \boldsymbol{x}_{\mathcal{R}}^t + \eta \boldsymbol{g}_{\mathcal{R}}^t\|^2$$

$$= \|\boldsymbol{x}_{\mathcal{I}\setminus\mathcal{R}}^{t+1} - \boldsymbol{x}_{\mathcal{I}\setminus\mathcal{R}}^t + \eta \boldsymbol{g}_{\mathcal{I}\setminus\mathcal{R}}^t\|^2. \tag{15}$$

Next, we bound the size of $\mathcal{I} \setminus \mathcal{R}$ as $|\mathcal{I} \setminus \mathcal{R}| \leq |\mathcal{I}^{t+1}| + |(\mathcal{I}^t \setminus \mathcal{I}^{t+1}) \setminus \mathcal{R}| + |\mathcal{I}^*| \leq k + |(\mathcal{I}^{t+1} \cap \mathcal{I}^*) \setminus \mathcal{I}^t| + k^* \leq k + 2k^*$. Also, since $\mathcal{I}^{t+1} \subseteq (\mathcal{I} \setminus \mathcal{R})$, we have $\boldsymbol{x}_{\mathcal{I}\setminus\mathcal{R}}^{t+1} = \Phi_k(\boldsymbol{x}_{\mathcal{I}\setminus\mathcal{R}}^t - \eta \boldsymbol{g}_{\mathcal{I}\setminus\mathcal{R}}^t)$. By combining Eqn. (15) and Lemma 2, we can obtain

$$\|\boldsymbol{x}_{\mathcal{I}}^{t+1} - \boldsymbol{x}_{\mathcal{I}}^t + \eta \boldsymbol{g}_{\mathcal{I}}^t\|^2 - \eta^2\|\boldsymbol{g}_{\mathcal{I}\setminus(\mathcal{I}^t \cup \mathcal{I}^*)}^t\|^2$$

$$\leq \frac{2k^*}{k + k^*}\|\boldsymbol{x}_{\mathcal{I}\setminus\mathcal{R}}^* - \boldsymbol{x}_{\mathcal{I}\setminus\mathcal{R}}^t + \eta \boldsymbol{g}_{\mathcal{I}\setminus\mathcal{R}}^t\|^2$$

$$\leq \frac{2k^*}{k + k^*}\|\boldsymbol{x}_{\mathcal{I}}^* - \boldsymbol{x}_{\mathcal{I}}^t + \eta \boldsymbol{g}_{\mathcal{I}}^t\|^2$$

$$= \frac{2k^*}{k + k^*}\left(\|\boldsymbol{x}^* - \boldsymbol{x}^t\|^2 + 2\eta\langle \boldsymbol{g}^t, \boldsymbol{x}^* - \boldsymbol{x}^t \rangle + \eta^2\|\boldsymbol{g}_{\mathcal{I}}^t\|^2\right)$$

$$\overset{\textcircled{1}}{=} \frac{2k^*}{k + k^*}\left(\|\boldsymbol{x}^* - \boldsymbol{x}^t\|^2 + 2\eta\langle \nabla f(\boldsymbol{x}^t), \boldsymbol{x}^* - \boldsymbol{x}^t \rangle + \eta^2\|\boldsymbol{g}_{\mathcal{I}}^t\|^2\right) + \xi\langle \boldsymbol{g}^t - \nabla f(\boldsymbol{x}^t), \boldsymbol{x}^* - \boldsymbol{x}^t \rangle$$

$$\leq \frac{2k^*}{k + k^*}\left[\|\boldsymbol{x}^* - \boldsymbol{x}^t\|^2 + 2\eta\left(f(\boldsymbol{x}^*) - f(\boldsymbol{x}^t) - \frac{\rho_s}{2}\|\boldsymbol{x}^* - \boldsymbol{x}^t\|^2\right) + \eta^2\|\boldsymbol{g}_{\mathcal{I}}^t\|^2\right] + \xi\langle \boldsymbol{g}^t - \nabla f(\boldsymbol{x}^t), \boldsymbol{x}^* - \boldsymbol{x}^t \rangle$$

$$= \frac{4\eta k^*}{k + k^*}(f(\boldsymbol{x}^*) - f(\boldsymbol{x}^t)) + \frac{2(1 - \eta\rho_s)k^*}{k + k^*}\|\boldsymbol{x}^* - \boldsymbol{x}^t\|^2 + \frac{2\eta^2 k^*}{k + k^*}\|\boldsymbol{g}_{\mathcal{I}}^t\|^2 + \xi\langle \boldsymbol{g}^t - \nabla f(\boldsymbol{x}^t), \boldsymbol{x}^* - \boldsymbol{x}^t \rangle$$

$$= \frac{4\eta k^*}{k + k^*}(f(\boldsymbol{x}^*) - f(\boldsymbol{x}^t)) + \frac{2(1 - \eta\rho_s)k^*}{k + k^*}\|\boldsymbol{x}^* - \boldsymbol{x}^t\|^2 + \frac{2\eta^2 k^*}{k + k^*}\|\boldsymbol{g}_{\mathcal{I}\setminus(\mathcal{I}^t \cup \mathcal{I}^*)}^t\|^2 + \frac{2\eta^2 k^*}{k + k^*}\|\boldsymbol{g}_{\mathcal{I}^t \cup \mathcal{I}^*}^t\|^2$$

$$\quad + \xi\langle \boldsymbol{g}^t - \nabla f(\boldsymbol{x}^t), \boldsymbol{x}^* - \boldsymbol{x}^t \rangle,$$

where $\xi = \frac{4\eta k^*}{k+k^*}$ in ①. By substituting the above inequality into Eqn. (13), we can further obtain

$$f(\boldsymbol{x}^{t+1})$$
$$\leq f(\boldsymbol{x}^t) + \frac{2k^*}{k+k^*}(f(\boldsymbol{x}^*) - f(\boldsymbol{x}^t)) + \frac{(1-\eta\rho_s)k^*}{\eta(k+k^*)}\|\boldsymbol{x}^* - \boldsymbol{x}^t\|^2 + \frac{\eta k^*}{k+k^*}\|\boldsymbol{g}_{\mathcal{I}\setminus(\mathcal{I}^t\cup\mathcal{I}^*)}^t\|^2$$
$$+ \left(\frac{\eta k^*}{k+k^*} - \frac{\eta}{2}\right)\|\boldsymbol{g}_{\mathcal{I}^t\cup\mathcal{I}^*}^t\|^2 - \frac{1-\eta\ell_s}{2\eta}\|\boldsymbol{x}^{t+1} - \boldsymbol{x}^t\|^2 + \frac{\xi}{2\eta}\langle\boldsymbol{g}^t - \nabla f(\boldsymbol{x}^t), \boldsymbol{x}^* - \boldsymbol{x}^t\rangle$$
$$+ \|\nabla f(\boldsymbol{x}^t) - \boldsymbol{g}^t\|\|\boldsymbol{x}^{t+1} - \boldsymbol{x}^t\|$$
$$\overset{①}{\leq} f(\boldsymbol{x}^t) + \frac{2k^*}{k+k^*}(f(\boldsymbol{x}^*) - f(\boldsymbol{x}^t)) + \frac{(1-\eta\rho_s)k^*}{\eta(k+k^*)}\|\boldsymbol{x}^* - \boldsymbol{x}^t\|^2 - \left[\frac{1-\eta\ell_s}{2\eta} - \frac{k^*}{\eta(k+k^*)}\right]\|\boldsymbol{x}^{t+1} - \boldsymbol{x}^t\|^2$$
$$+ \left(\frac{\eta k^*}{k+k^*} - \frac{\eta}{2}\right)\|\boldsymbol{g}_{\mathcal{I}^t\cup\mathcal{I}^*}^t\|^2 + \frac{\xi}{2\eta}\langle\boldsymbol{g}^t - \nabla f(\boldsymbol{x}^t), \boldsymbol{x}^* - \boldsymbol{x}^t\rangle + \|\nabla f(\boldsymbol{x}^t) - \boldsymbol{g}^t\|\|\boldsymbol{x}^{t+1} - \boldsymbol{x}^t\|$$
$$\overset{②}{\leq} f(\boldsymbol{x}^t) + \frac{2k^*}{k+k^*}(f(\boldsymbol{x}^*) - f(\boldsymbol{x}^t)) + \frac{(1-\eta\rho_s)k^*}{\eta(k+k^*)}\|\boldsymbol{x}^* - \boldsymbol{x}^t\|^2 - \left(\frac{\eta}{2} - \frac{\eta k^*}{k+k^*}\right)\|\boldsymbol{g}_{\mathcal{I}^t\cup\mathcal{I}^*}^t\|^2$$
$$+ \frac{\xi}{2\eta}\langle\boldsymbol{g}^t - \nabla f(\boldsymbol{x}^t), \boldsymbol{x}^* - \boldsymbol{x}^t\rangle + \frac{\eta(k+k^*)}{2\left((1-\eta\ell_s)k - (1+\eta\ell_s)k^*\right)}\|\nabla f(\boldsymbol{x}^t) - \boldsymbol{g}^t\|^2,$$

where in ① we used $\|\boldsymbol{x}^{t+1} - \boldsymbol{x}^t\| \geq \eta\|\boldsymbol{g}_{\mathcal{I}\setminus(\mathcal{I}^t\cup\mathcal{I}^*)}^t\|$, and in ② we have used the basic inequality $ab \leq \frac{a^2}{4c} + cb^2, \forall c > 0$. By invoking Lemma 5 in the above we further get

$$f(\boldsymbol{x}^{t+1})$$
$$\leq f(\boldsymbol{x}^t) + \frac{2k^*}{k+k^*}\left(1 + \frac{2(1-\eta\rho_s)}{\eta\rho_s}\right)(f(\boldsymbol{x}^*) - f(\boldsymbol{x}^t)) - \left(\frac{\eta}{2} - \frac{(\eta^2\rho_s^2 + 8(1-\eta\rho_s))k^*}{\eta\rho_s^2(k+k^*)}\right)\|\boldsymbol{g}_{\mathcal{I}^t\cup\mathcal{I}^*}^t\|^2$$
$$+ \frac{\xi}{2\eta}\langle\boldsymbol{g}^t - \nabla f(\boldsymbol{x}^t), \boldsymbol{x}^* - \boldsymbol{x}^t\rangle + \left(\frac{\eta(k+k^*)}{2\left((1-\eta\ell_s)k - (1+\eta\ell_s)k^*\right)} + \frac{8(1-\eta\rho_s)k^*}{\eta\rho_s^2(k+k^*)}\right)\|\nabla f(\boldsymbol{x}^t) - \boldsymbol{g}^t\|^2.$$

Let us now consider $\eta = \frac{1}{2\ell_s}$ in the above inequality, which leads to

$$f(\boldsymbol{x}^{t+1}) \leq f(\boldsymbol{x}^t) + \frac{2(4\ell_s - \rho_s)k^*}{\rho_s(k+k^*)}(f(\boldsymbol{x}^*) - f(\boldsymbol{x}^t)) - \left(\frac{1}{4\ell_s} - \frac{(\rho_s^2 - 16\rho_s\ell_s + 32\ell_s^2)k^*}{2\ell_s\rho_s^2(k+k^*)}\right)\|\boldsymbol{g}_{\mathcal{I}^t\cup\mathcal{I}^*}^t\|^2$$
$$+ \ell_s\xi\langle\boldsymbol{g}^t - \nabla f(\boldsymbol{x}^t), \boldsymbol{x}^* - \boldsymbol{x}^t\rangle + \left(\frac{k+k^*}{2\ell_s(k-3k^*)} + \frac{8(2\ell_s - \rho_s)k^*}{\rho_s^2(k+k^*)}\right)\|\nabla f(\boldsymbol{x}^t) - \boldsymbol{g}^t\|^2.$$

Since $k \geq \left(1 + \frac{64\ell_s^2}{\rho_s^2}\right)k^*$, with algebra manipulation we can further show that

$$f(\boldsymbol{x}^{t+1}) \leq f(\boldsymbol{x}^t) + \frac{\rho_s}{8\ell_s}(f(\boldsymbol{x}^*) - f(\boldsymbol{x}^t)) + \ell_s\xi\langle\boldsymbol{g}^t - \nabla f(\boldsymbol{x}^t), \boldsymbol{x}^* - \boldsymbol{x}^t\rangle + \left(\frac{5}{4\ell_s} + \frac{8\ell_s}{\rho_s^2}\right)\|\nabla f(\boldsymbol{x}^t) - \boldsymbol{g}^t\|^2$$
$$\leq f(\boldsymbol{x}^t) + \frac{\rho_s}{8\ell_s}(f(\boldsymbol{x}^*) - f(\boldsymbol{x}^t)) + \ell_s\xi\langle\boldsymbol{g}^t - \nabla f(\boldsymbol{x}^t), \boldsymbol{x}^* - \boldsymbol{x}^t\rangle + \frac{37\ell_s}{4\rho_s^2}\|\nabla f(\boldsymbol{x}^t) - \boldsymbol{g}^t\|^2.$$

Taking expectation (conditioned on $\boldsymbol{x}^t$) on both sides of the above we arrive at

$$\mathbb{E}[f(\boldsymbol{x}^{t+1}) \mid \boldsymbol{x}^t] \leq f(\boldsymbol{x}^t) + \frac{\rho_s}{8\ell_s}(f(\boldsymbol{x}^*) - f(\boldsymbol{x}^t)) + \frac{37\ell_s}{4\rho_s^2}\mathbb{E}[\|\nabla f(\boldsymbol{x}^t) - \boldsymbol{g}^t\|^2 \mid \boldsymbol{x}^t]$$
$$\overset{①}{\leq} f(\boldsymbol{x}^t) + \frac{\rho_s}{8\ell_s}(f(\boldsymbol{x}^*) - f(\boldsymbol{x}^t)) + \frac{37\ell_s}{4\rho_s^2 s_t}B,$$

where ① uses Lemma 3, in which $B = \max_t B_t$ and $B_t = \frac{1}{n-1}\sum_{i=1}^n\|\nabla f_i(\boldsymbol{x}^t) - \nabla f(\boldsymbol{x}^t)\|^2$. By further taking expectation on $\boldsymbol{x}^t$ we obtain

$$\mathbb{E}[f(\boldsymbol{x}^{t+1}) - f(\boldsymbol{x}^*)] \leq \left(1 - \frac{\rho_s}{8\ell_s}\right)\mathbb{E}[f(\boldsymbol{x}^t) - f(\boldsymbol{x}^*)] + \frac{37\ell_s}{4\rho_s^2 s_t}B.$$

We further set $\beta = 1 - \frac{1}{8\kappa_s}$ and $s_t = \tau/\omega^t$, and then assume that $\tau$ is large enough such that

$$\gamma := \frac{37\ell_s B}{4\tau\rho_s^2} \leq \delta[f(\boldsymbol{x}^0) - f(\boldsymbol{x}^*)], \tag{16}$$

where $\delta$ is a positive constant and will be discussed later. Now we use mathematical induction to prove

$$\mathbb{E}[f(\boldsymbol{x}^t) - f(\boldsymbol{x}^*)] \leq \theta^t[f(\boldsymbol{x}^0) - f(\boldsymbol{x}^*)], \ (\forall t), \tag{17}$$

where $\theta < 1$ is a constant and will be given below.

Obviously, when $t = 0$, Eqn. (17) holds. Now assume that for all $k \leq t$, Eqn. (17) holds. Then for $k = t + 1$, we have

$$\mathbb{E}[f(\boldsymbol{x}^{t+1}) - f(\boldsymbol{x}^*)] \leq \beta\mathbb{E}[f(\boldsymbol{x}^t) - f(\boldsymbol{x}^*)] + \delta\omega^t[f(\boldsymbol{x}^0) - f(\boldsymbol{x}^*)]$$

$$\leq (\beta\theta^t + \delta\omega^t)[f(\boldsymbol{x}^0) - f(\boldsymbol{x}^*)] \overset{①}{\leq} \theta^{t+1}[f(\boldsymbol{x}^0) - f(\boldsymbol{x}^*)],$$

where ① holds since we let

$$\omega = \theta = \beta + \delta. \tag{18}$$

This means that if Eqn. (16) holds, then Eqn. (17) always holds. So the conclusion holds.

Then let $\tau \geq \frac{148B\kappa_s^2}{\rho_s[f(\boldsymbol{x}^0) - f(\boldsymbol{x}^*)]}$ which gives $\delta \leq \frac{1}{16\kappa_s}$. We have

$$\omega = \theta = \beta + \delta \leq 1 - \frac{1}{8\kappa_s} + \frac{1}{16\kappa_s} = 1 - \frac{1}{16\kappa_s}.$$

Therefore, we have

$$\mathbb{E}[f(\boldsymbol{x}^t) - f(\boldsymbol{x}^*)] \leq \left(1 - \frac{1}{16\kappa_s}\right)^t [f(\boldsymbol{x}^0) - f(\boldsymbol{x}^*)].$$

The proof is completed. $\qquad\square$

## C   Proofs for Section 4

### C.1   Proofs of Theorem 3

*Proof.* Here we also first consider the sparsity-constrained problem. Let us consider $\mathcal{I} = \mathcal{I}^{t+1} \cup \mathcal{I}^t \cup \mathcal{I}^{t-1} \cup \mathcal{I}^*$ and $\boldsymbol{y}^{t+1} = \boldsymbol{x}^t - \eta\boldsymbol{g}_{\mathcal{I}}^t + \nu(\boldsymbol{x}^t - \boldsymbol{x}^{t-1})$. Since $f(\boldsymbol{x})$ is quadratic with Hessian $\boldsymbol{H}$, we know that

$$\boldsymbol{y}_{\mathcal{I}}^{t+1} - \boldsymbol{x}^* = \boldsymbol{x}^t - \boldsymbol{x}^* - \eta\nabla_{\mathcal{I}}f(\boldsymbol{x}^t) + \eta\nabla_{\mathcal{I}}f(\boldsymbol{x}^*) + \nu(\boldsymbol{x}^t - \boldsymbol{x}^{t-1}) + \eta(\nabla_{\mathcal{I}}f(\boldsymbol{x}^t) - \boldsymbol{g}_{\mathcal{I}}^t) - \eta\nabla_{\mathcal{I}}f(\boldsymbol{x}^*)$$

$$= ((1+\nu)\boldsymbol{I} - \eta\boldsymbol{H}_{\mathcal{I}\mathcal{I}})(\boldsymbol{x}^t - \boldsymbol{x}^*) - \nu(\boldsymbol{x}^{t-1} - \boldsymbol{x}^*) + \eta(\nabla_{\mathcal{I}}f(\boldsymbol{x}^t) - \boldsymbol{g}_{\mathcal{I}}^t) - \eta\nabla_{\mathcal{I}}f(\boldsymbol{x}^*). \tag{19}$$

Since the above iterate depends on the previous two iterates, we consider the following three-term recurrence in matrix form:

$$\begin{bmatrix} \boldsymbol{y}_{\mathcal{I}}^{t+1} - \boldsymbol{x}^* \\ \boldsymbol{x}^t - \boldsymbol{x}^* \end{bmatrix} = \begin{bmatrix} (1+\nu)\boldsymbol{I} - \eta\boldsymbol{H}_{\mathcal{I}\mathcal{I}} & -\nu\boldsymbol{I} \\ \boldsymbol{I} & 0 \end{bmatrix} \begin{bmatrix} \boldsymbol{x}^t - \boldsymbol{x}^* \\ \boldsymbol{x}^{t-1} - \boldsymbol{x}^* \end{bmatrix} + \eta \begin{bmatrix} \nabla_{\mathcal{I}}f(\boldsymbol{x}^t) - \boldsymbol{g}_{\mathcal{I}}^t - \nabla_{\mathcal{I}}f(\boldsymbol{x}^*) \\ 0 \end{bmatrix}.$$

Since $\boldsymbol{x}^{t+1} = \Phi_k(\boldsymbol{y}^{t+1}) = \Phi_k(\boldsymbol{y}_{\mathcal{I}}^{t+1})$, based on Lemma 1 we get $\|\boldsymbol{x}^{t+1} - \boldsymbol{x}^*\| \leq \sqrt{\alpha}\|\boldsymbol{y}_{\mathcal{I}}^{t+1} - \boldsymbol{x}^*\| \leq \alpha\|\boldsymbol{y}_{\mathcal{I}}^{t+1} - \boldsymbol{x}^*\|$, where $\alpha = 1 + \frac{2\sqrt{k^*}}{\sqrt{k-k^*}}$. Let $\widehat{\mathcal{I}} = \mathsf{supp}(\Phi_{3k}(\nabla f(\boldsymbol{x}^*))) \cup \mathsf{supp}(\boldsymbol{x}^*)$ and $\widehat{s} = 3k + k^*$. Then

$$
\mathbb{E}\left\|\begin{bmatrix} \boldsymbol{x}^{t+1} - \boldsymbol{x}^* \\ \boldsymbol{x}^t - \boldsymbol{x}^* \end{bmatrix}\right\|
$$
$$
\leq \alpha\mathbb{E}\left\|\begin{bmatrix} \boldsymbol{y}_{\mathcal{I}}^{t+1} - \boldsymbol{x}^* \\ \boldsymbol{x}^t - \boldsymbol{x}^* \end{bmatrix}\right\|
$$
$$
\leq \alpha\mathbb{E}\left\|\begin{bmatrix} (1+\nu)\boldsymbol{I} - \eta\boldsymbol{H}_{\mathcal{I}\mathcal{I}} & -\nu\boldsymbol{I} \\ \boldsymbol{I} & 0 \end{bmatrix}\right\|\left\|\begin{bmatrix} \boldsymbol{x}^t - \boldsymbol{x}^* \\ \boldsymbol{x}^{t-1} - \boldsymbol{x}^* \end{bmatrix}\right\| + \alpha\eta\mathbb{E}\left\|\begin{bmatrix} \nabla_{\mathcal{I}}f(\boldsymbol{x}^t) - \boldsymbol{g}_{\mathcal{I}}^t - \nabla_{\mathcal{I}}f(\boldsymbol{x}^*) \\ 0 \end{bmatrix}\right\|
$$
$$
\leq \alpha\mathbb{E}\left\|\begin{bmatrix} (1+\nu)\boldsymbol{I}_{\mathcal{I}\mathcal{I}} - \eta\boldsymbol{H}_{\mathcal{I}\mathcal{I}} & -\nu\boldsymbol{I}_{\mathcal{I}\mathcal{I}} \\ \boldsymbol{I}_{\mathcal{I}\mathcal{I}} & 0 \end{bmatrix}\right\|\left\|\begin{bmatrix} \boldsymbol{x}^t - \boldsymbol{x}^* \\ \boldsymbol{x}^{t-1} - \boldsymbol{x}^* \end{bmatrix}\right\| + \alpha\eta\mathbb{E}\|\nabla_{\mathcal{I}}f(\boldsymbol{x}^t) - \boldsymbol{g}_{\mathcal{I}}^t\| + \eta\|\nabla_{\mathcal{I}}f(\boldsymbol{x}^*)\|
$$
$$
\overset{①}{\leq} \alpha\frac{\sqrt{\ell_{\widehat{s}}} - \sqrt{\rho_{\widehat{s}}}}{\sqrt{\ell_{\widehat{s}}} + \sqrt{\rho_{\widehat{s}}}}\mathbb{E}\left\|\begin{bmatrix} \boldsymbol{x}^t - \boldsymbol{x}^* \\ \boldsymbol{x}^{t-1} - \boldsymbol{x}^* \end{bmatrix}\right\| + \alpha\eta\mathbb{E}\|\nabla_{\mathcal{I}}f(\boldsymbol{x}^t) - \boldsymbol{g}_{\mathcal{I}}^t\| + \alpha\eta\|\nabla_{\mathcal{I}}f(\boldsymbol{x}^*)\|
$$
$$
\overset{②}{\leq} \alpha\left(1 - \sqrt{\frac{\rho_{\widehat{s}}}{\ell_{\widehat{s}}}}\right)\mathbb{E}\left\|\begin{bmatrix} \boldsymbol{x}^t - \boldsymbol{x}^* \\ \boldsymbol{x}^{t-1} - \boldsymbol{x}^* \end{bmatrix}\right\| + \alpha\eta\mathbb{E}\|\nabla_{\mathcal{I}}f(\boldsymbol{x}^t) - \boldsymbol{g}_{\mathcal{I}}^t\| + \alpha\eta\|\nabla_{\mathcal{I}}f(\boldsymbol{x}^*)\|
$$
$$
\overset{③}{\leq} \left(1 - \frac{1}{2}\sqrt{\frac{\rho_{\widehat{s}}}{\ell_{\widehat{s}}}}\right)\mathbb{E}\left\|\begin{bmatrix} \boldsymbol{x}^t - \boldsymbol{x}^* \\ \boldsymbol{x}^{t-1} - \boldsymbol{x}^* \end{bmatrix}\right\| + \alpha\eta\mathbb{E}\|\nabla_{\mathcal{I}}f(\boldsymbol{x}^t) - \boldsymbol{g}_{\mathcal{I}}^t\| + \alpha\eta\|\nabla_{\mathcal{I}}f(\boldsymbol{x}^*)\|
$$
$$
\overset{④}{\leq} \left(1 - \frac{1}{2}\sqrt{\frac{\rho_{\widehat{s}}}{\ell_{\widehat{s}}}}\right)\mathbb{E}\left\|\begin{bmatrix} \boldsymbol{x}^t - \boldsymbol{x}^* \\ \boldsymbol{x}^{t-1} - \boldsymbol{x}^* \end{bmatrix}\right\| + \alpha\eta\sqrt{\frac{B}{s_t}} + \alpha\eta\|\nabla_{\widehat{\mathcal{I}}}f(\boldsymbol{x}^*)\|,
$$
$$(20)$$

where ① follows from Lemma 6 with $\eta = \frac{4}{(\sqrt{\rho_{\widehat{s}}} + \sqrt{\ell_{\widehat{s}}})^2}$ and $\nu = \max\left\{|1 - \sqrt{\eta\rho_{\widehat{s}}}|^2, |1 - \sqrt{\eta\ell_{\widehat{s}}}|^2\right\}$ (For the sake of readability, we have assumed without loss of generality that $\rho\left(\begin{bmatrix} (1+\nu)\boldsymbol{I}_{\mathcal{I}\mathcal{I}} - \eta\boldsymbol{H}_{\mathcal{I}\mathcal{I}} & -\nu\boldsymbol{I}_{\mathcal{I}\mathcal{I}} \\ \boldsymbol{I}_{\mathcal{I}\mathcal{I}} & 0 \end{bmatrix}\right) = \left\|\begin{bmatrix} (1+\nu)\boldsymbol{I}_{\mathcal{I}\mathcal{I}} - \eta\boldsymbol{H}_{\mathcal{I}\mathcal{I}} & -\nu\boldsymbol{I}_{\mathcal{I}\mathcal{I}} \\ \boldsymbol{I}_{\mathcal{I}\mathcal{I}} & 0 \end{bmatrix}\right\|$. Otherwise, we may simply run sufficient rounds of heavy-ball iteration over $\mathcal{I}$ before applying the hard thresholding operation and similar complexity bounds can be obtained with more involved arguments), ② follows from the fact $\ell_{\widehat{s}} \geq \rho_{\widehat{s}}$, ③ follows from the condition $k \geq \left(1 + \frac{16\ell_{\widehat{s}}}{\rho_{\widehat{s}}}\right)k^*$ which implies $\alpha \leq 1 + \frac{1}{2}\sqrt{\frac{\rho_{\widehat{s}}}{\ell_{\widehat{s}}}}$, ④ uses Lemma 3 with $B = \max_t B_t$ and $\|\nabla_{\mathcal{I}}f(\boldsymbol{x}^*)\| \leq \|\nabla_{\widehat{\mathcal{I}}}f(\boldsymbol{x}^*)\|$ where $\widehat{\mathcal{I}} = \mathsf{supp}(\Phi_{3k}(\nabla f(\boldsymbol{x}^*))) \cup \mathsf{supp}(\boldsymbol{x}^*)$.

Now we let

$$
\beta := 1 - \frac{1}{2\sqrt{\kappa_s}} \quad \text{where} \quad \kappa_s = \frac{\ell_s}{\rho_s}.
$$

We further set $s_t = \tau/\omega^t$ and assume that $\tau$ is large enough such that

$$
\gamma := \frac{\alpha\eta\sqrt{B}}{\sqrt{\tau}} \leq \frac{8\sqrt{B}}{\sqrt{\tau}(\sqrt{\rho_{\widehat{s}}} + \sqrt{\ell_{\widehat{s}}})^2} \leq \delta(\|\boldsymbol{x}^0 - \boldsymbol{x}^*\| + \|\boldsymbol{x}^{-1} - \boldsymbol{x}^*\|), \quad (21)
$$

where $\delta$ is a positive constant and will be discussed later. Now we use mathematical induction to prove

$$
\mathbb{E}\left\|\begin{bmatrix} \boldsymbol{x}^t - \boldsymbol{x}^* \\ \boldsymbol{x}^{t-1} - \boldsymbol{x}^* \end{bmatrix}\right\| \leq \theta^t\left\|\begin{bmatrix} \boldsymbol{x}^0 - \boldsymbol{x}^* \\ \boldsymbol{x}^{-1} - \boldsymbol{x}^* \end{bmatrix}\right\| + \frac{8}{(1-\beta)(\sqrt{\rho_{\widehat{s}}} + \sqrt{\ell_{\widehat{s}}})^2}\|\nabla_{\widehat{\mathcal{I}}}f(\boldsymbol{x}^*)\|, \ (\forall t), \quad (22)
$$

where $\theta < 1$ is a constant and will be given below.

Obviously, when $t = 0$, Eqn. (22) holds. Now assume that for all $k \leq t$, Eqn. (22) holds. Then for $k = t + 1$, we have

$$\mathbb{E}\left\|\begin{bmatrix} \boldsymbol{x}^t - \boldsymbol{x}^* \\ \boldsymbol{x}^{t-1} - \boldsymbol{x}^* \end{bmatrix}\right\| \leq \beta\mathbb{E}\left\|\begin{bmatrix} \boldsymbol{x}^{t-1} - \boldsymbol{x}^* \\ \boldsymbol{x}^{t-2} - \boldsymbol{x}^* \end{bmatrix}\right\| + \delta\omega^{\frac{t}{2}}\left\|\begin{bmatrix} \boldsymbol{x}^0 - \boldsymbol{x}^* \\ \boldsymbol{x}^{-1} - \boldsymbol{x}^* \end{bmatrix}\right\| + \frac{8}{(\sqrt{\rho_{\widehat{s}}} + \sqrt{\ell_{\widehat{s}}})^2}\|\nabla_{\widehat{\mathcal{I}}}f(\boldsymbol{x}^*)\|$$

$$\leq (\beta\theta^t + \delta\omega^{\frac{t}{2}})\mathbb{E}\left\|\begin{bmatrix} \boldsymbol{x}^0 - \boldsymbol{x}^* \\ \boldsymbol{x}^{-1} - \boldsymbol{x}^* \end{bmatrix}\right\| + \frac{8}{(\sqrt{\rho_{\widehat{s}}} + \sqrt{\ell_{\widehat{s}}})^2}\left[\frac{\beta}{1-\beta} + 1\right]\|\nabla_{\widehat{\mathcal{I}}}f(\boldsymbol{x}^*)\|$$

$$\overset{①}{\leq} \theta^t\mathbb{E}\left\|\begin{bmatrix} \boldsymbol{x}^0 - \boldsymbol{x}^* \\ \boldsymbol{x}^{-1} - \boldsymbol{x}^* \end{bmatrix}\right\| + \frac{8}{(1-\beta)(\sqrt{\rho_{\widehat{s}}} + \sqrt{\ell_{\widehat{s}}})^2}\|\nabla_{\widehat{\mathcal{I}}}f(\boldsymbol{x}^*)\|$$

where ① holds since we let

$$\omega = \theta^2 \quad \text{and} \quad \theta = \beta + \delta. \tag{23}$$

This means that if Eqn. (23) holds, then Eqn. (22) always holds. So the conclusion holds.

Then let $\tau \geq \frac{81B\kappa_s}{4(\sqrt{\rho_s} + \sqrt{\ell_s})^4\|\boldsymbol{x}^0 - \boldsymbol{x}^*\|^2}$. Recall $\boldsymbol{x}^{-1} = \boldsymbol{x}^0$, we have

$$\theta = \beta + \delta \leq 1 - \frac{1}{2\sqrt{\kappa_s}} + \frac{4}{9\sqrt{\kappa_s}} = 1 - \frac{1}{18\sqrt{\kappa_s}}.$$

Therefore, we have

$$\mathbb{E}\|\boldsymbol{x}^t - \boldsymbol{x}^*\| \leq 2\theta^t\|\boldsymbol{x}^0 - \boldsymbol{x}^*\| + \frac{8}{(1-\beta)(\sqrt{\rho_{\widehat{s}}} + \sqrt{\ell_{\widehat{s}}})^2}\|\nabla_{\widehat{\mathcal{I}}}f(\boldsymbol{x}^*)\|$$

$$= 2\theta^t\|\boldsymbol{x}^0 - \boldsymbol{x}^*\| + \frac{16\sqrt{\kappa_{\widehat{s}}}}{(\sqrt{\rho_{\widehat{s}}} + \sqrt{\ell_{\widehat{s}}})^2}\|\nabla_{\widehat{\mathcal{I}}}f(\boldsymbol{x}^*)\|, \ (\forall t)$$

where $\beta = 1 - \frac{1}{2\sqrt{\kappa_s}}$ and $\theta \leq 1 - \frac{1}{18\sqrt{\kappa_s}}$ with $\omega = \theta^2$. So the result on sparsity-constrained problem holds.

Now we consider the rank-constrained problem. Since the proof accesses the Hessian, here we need to vectorize the matrix variable $\boldsymbol{x}$. For notation simplicity, we use $\widetilde{\boldsymbol{x}} \in \mathbb{R}^{d_1d_2}$ to denote the vectorization of $\boldsymbol{x} \in \mathbb{R}^{d_1 \times d_2}$. Assume that the skinny SVDs of $\boldsymbol{x}^t$, and $\boldsymbol{x}^*$ are respectively $\boldsymbol{x}^t = \boldsymbol{U}_t\boldsymbol{\Sigma}_t\boldsymbol{V}_t^T$ and $\boldsymbol{x}^* = \boldsymbol{U}_*\boldsymbol{\Sigma}_*\boldsymbol{V}_*^T$. Then we have $\boldsymbol{x}^t = \boldsymbol{U}_t\boldsymbol{U}_t^T\boldsymbol{x}^t$ and $\boldsymbol{x}^* = \boldsymbol{U}_*\boldsymbol{U}_*^T\boldsymbol{x}^*$. We further define the projection operation $\mathcal{P}_{\boldsymbol{U}}(\boldsymbol{x}) = \boldsymbol{U}\boldsymbol{U}^T\boldsymbol{x}$ which projects $\boldsymbol{x}$ in the subspace spanned by $\boldsymbol{U}$. Then let $\mathcal{I} = \mathcal{I}^* \cup \mathcal{I}^{t-1} \cup \mathcal{I}^t \cup \mathcal{I}^{t+1}$, where $\mathcal{I}^* = \mathsf{supp}(\boldsymbol{x}^*)$, $\mathcal{I}^{t-1} = \mathsf{supp}(\boldsymbol{x}^{t-1})$, $\mathcal{I}^t = \mathsf{supp}(\boldsymbol{x}^t)$ and $\mathcal{I}^{t+1} = \mathsf{supp}(\boldsymbol{x}^{t+1})$. The notation $\boldsymbol{x}^t_{\mathcal{I}^t} = \boldsymbol{U}_t\boldsymbol{U}_t^T\boldsymbol{x}^t = \mathcal{P}_{\boldsymbol{U}^t}(\boldsymbol{x}^t)$. For the specifical notation $\boldsymbol{y}_{\mathcal{I}}$, it denotes $\boldsymbol{y}_{\mathcal{I}} = \mathcal{P}_{\mathcal{I}}(\boldsymbol{y}_{\mathcal{I}}) = (\boldsymbol{U}_{t-1}\boldsymbol{U}_{t-1}^T + \boldsymbol{U}_t\boldsymbol{U}_t^T + \boldsymbol{U}_{t+1}\boldsymbol{U}_{t+1}^T + \boldsymbol{U}_*\boldsymbol{U}_*^T - \boldsymbol{U}_{t-1}\boldsymbol{U}_{t-1}^T\boldsymbol{U}_t\boldsymbol{U}_t^T\boldsymbol{U}_{t+1}\boldsymbol{U}_{t+1}^T - \boldsymbol{U}_{t-1}\boldsymbol{U}_{t-1}^T\boldsymbol{U}_t\boldsymbol{U}_t^T\boldsymbol{U}_*\boldsymbol{U}_*^T - \boldsymbol{U}_{t-1}\boldsymbol{U}_{t-1}^T\boldsymbol{U}_{t+1}\boldsymbol{U}_{t+1}^T\boldsymbol{U}_*\boldsymbol{U}_*^T - \boldsymbol{U}_t\boldsymbol{U}_t^T\boldsymbol{U}_{t+1}\boldsymbol{U}_{t+1}^T\boldsymbol{U}_*\boldsymbol{U}_*^T + \boldsymbol{U}_{t-1}\boldsymbol{U}_{t-1}^T\boldsymbol{U}_t\boldsymbol{U}_t^T\boldsymbol{U}_{t+1}\boldsymbol{U}_{t+1}^T\boldsymbol{U}_*\boldsymbol{U}_*^T)\boldsymbol{g}^t$ where $\mathcal{P}_{\mathcal{I}}$ is a projection. Here $\boldsymbol{y}$ can be $\boldsymbol{x}$, $\boldsymbol{g}^t$ and $\nabla f(\boldsymbol{x})$.

Then we also prove Eqn. (19) holds. Let $\widetilde{\boldsymbol{y}}^t$, $\widetilde{\boldsymbol{x}}^*$, $\widetilde{\boldsymbol{x}}^t$ and $\nabla\widetilde{f}(\boldsymbol{x})$ respectively denotes the vectorization of $\boldsymbol{y}^t$, $\boldsymbol{x}^*$, $\boldsymbol{x}^t$ and $\nabla f(\boldsymbol{x})$. The notation $\widetilde{\boldsymbol{x}}_{\mathcal{I}}$ denotes the vectorization of $\boldsymbol{U}_{\mathcal{I}}\boldsymbol{U}_{\mathcal{I}}^T\widetilde{\boldsymbol{x}}$. Then we have

$$\widetilde{\boldsymbol{y}}^{t+1}_{\mathcal{I}} - \widetilde{\boldsymbol{x}}^* = \widetilde{\boldsymbol{x}}^t - \widetilde{\boldsymbol{x}}^* - \eta\nabla_{\mathcal{I}}\widetilde{f}(\boldsymbol{x}^t) + \eta\nabla_{\mathcal{I}}\widetilde{f}(\boldsymbol{x}^*) + \nu(\widetilde{\boldsymbol{x}}^t - \widetilde{\boldsymbol{x}}^{t-1}) + \eta(\nabla_{\mathcal{I}}\widetilde{f}(\boldsymbol{x}^t) - \widetilde{\boldsymbol{g}}^t_{\mathcal{I}}) - \eta\nabla_{\mathcal{I}}\widetilde{f}(\boldsymbol{x}^*)$$

$$= ((1+\nu)\boldsymbol{I} - \eta\boldsymbol{H})(\widetilde{\boldsymbol{x}}^t - \widetilde{\boldsymbol{x}}^*) - \nu(\widetilde{\boldsymbol{x}}^{t-1} - \widetilde{\boldsymbol{x}}^*) + \eta(\nabla_{\mathcal{I}}\widetilde{f}(\widetilde{\boldsymbol{x}}^t) - \widetilde{\boldsymbol{g}}^t_{\mathcal{I}}) - \eta\nabla_{\mathcal{I}}\widetilde{f}(\boldsymbol{x}^*),$$

where $\boldsymbol{H} = \mathcal{P}_{\mathcal{I}}(\nabla^2\widetilde{f}(\widetilde{\boldsymbol{z}}))$. Here $\nabla^2\widetilde{f}(\widetilde{\boldsymbol{z}})$ comes from the fact that $\nabla_{\mathcal{I}}\widetilde{f}(\boldsymbol{x}^t) - \nabla_{\mathcal{I}}\widetilde{f}(\boldsymbol{x}^*) = \mathcal{P}_{\mathcal{I}}(\nabla\widetilde{f}(\boldsymbol{x}^t) - \nabla\widetilde{f}(\boldsymbol{x}^*)) \overset{①}{=} \mathcal{P}_{\mathcal{I}}\nabla^2\widetilde{f}(\widetilde{\boldsymbol{z}})(\widetilde{\boldsymbol{x}}^t - \widetilde{\boldsymbol{x}}^*)$ in which ① uses the second differential property of $f(\boldsymbol{x})$ and thus there exists a matrix $\boldsymbol{z}$ such that $\nabla\widetilde{f}(\boldsymbol{x}^t) - \nabla\widetilde{f}(\boldsymbol{x}^*) = \nabla^2\widetilde{f}(\widetilde{\boldsymbol{z}})(\widetilde{\boldsymbol{x}}^t - \widetilde{\boldsymbol{x}}^*)$. Then by Assumptions 1 and 2, we have $\rho_{\widehat{s}}\boldsymbol{I} \preceq \mathcal{P}_{\mathcal{I}}(\boldsymbol{H}) \preceq \ell_{\widehat{s}}\boldsymbol{I}$ since $\|\mathcal{P}_{\mathcal{I}}\|_2 \leq 1$. We obtain $\|\mathcal{P}_{\mathcal{I}}\|_2 \leq 1$ since we have $\mathcal{P}_{\mathcal{I}}^T\mathcal{P}_{\mathcal{I}} = \mathcal{P}_{\mathcal{I}}$.

On the other hand, we can follow Eqn. (12) in Section B.1 to prove

$$\Phi_k(\boldsymbol{y}^{t+1}_{\mathcal{I}}) = \boldsymbol{U}_{t+1}\boldsymbol{U}_{t+1}^T[\boldsymbol{x}^t - \eta\boldsymbol{g}^t_{\mathcal{I}} + \nu(\boldsymbol{x}^t - \boldsymbol{x}^{t-1})]$$

$$= \boldsymbol{U}_{t+1}\boldsymbol{U}_{t+1}^T[\boldsymbol{x}^t + \nu(\boldsymbol{x}^t - \boldsymbol{x}^{t-1})] + \eta\boldsymbol{U}_{t+1}\boldsymbol{U}_{t+1}^T\mathcal{P}_{\mathcal{I}}(\boldsymbol{g}^t)$$

$$\overset{①}{=} \boldsymbol{U}_{t+1}\boldsymbol{U}_{t+1}^T[\boldsymbol{x}^t + \nu(\boldsymbol{x}^t - \boldsymbol{x}^{t-1})] + \eta\boldsymbol{U}_{t+1}\boldsymbol{U}_{t+1}^T\boldsymbol{g}^t = \boldsymbol{x}^{t+1},$$

where ① plugs $\mathcal{P}_\mathcal{I}$ defined above and obtains $\boldsymbol{U}_{t+1}\boldsymbol{U}_{t+1}^T\mathcal{P}_\mathcal{I} = \boldsymbol{U}_{t+1}\boldsymbol{U}_{t+1}^T$. Then we apply Lemma 1 to establish

$$\|\boldsymbol{x}^{t+1} - \boldsymbol{x}^*\|^2 = \|\Phi_k(\boldsymbol{y}_\mathcal{I}^{t+1}) - \boldsymbol{x}^*\|^2 \leq \left(1 + \frac{2\sqrt{k^*}}{\sqrt{k - k^*}}\right)\|\boldsymbol{y}_\mathcal{I}^{t+1} - \boldsymbol{x}^*\|^2,$$

where $\alpha = 1 + \frac{2\sqrt{k^*}}{\sqrt{k-k^*}}$. Therefore, we can establish

$$\mathbb{E}\left\|\begin{bmatrix} \boldsymbol{x}^{t+1} - \boldsymbol{x}^* \\ \boldsymbol{x}^t - \boldsymbol{x}^* \end{bmatrix}\right\| \leq \alpha\mathbb{E}\left\|\begin{bmatrix} \boldsymbol{y}_\mathcal{I}^{t+1} - \boldsymbol{x}^* \\ \boldsymbol{x}^t - \boldsymbol{x}^* \end{bmatrix}\right\|.$$

Then we can establish the first inequality in Eqn. (20). The following proof does not depend the property on rank-constrained problem. So we can just follow the above proof sketch for sparsity-constrained problem to prove the result on the rank-constrained problem. The proof is completed. □

### C.2 Proof of Corollary 2

*Proof.* To achieve $\epsilon$-accurate solution, let

$$2\theta^t\|\boldsymbol{x}^0 - \boldsymbol{x}^*\| \leq \sqrt{\epsilon}$$

where $\widetilde{\beta} = 1 - \frac{1}{2\sqrt{\kappa_{\widehat{s}}}}$ and $\theta \leq 1 - \frac{1}{18\sqrt{\kappa_{\widehat{s}}}}$ with $\omega = \theta^2$, we have

$$T \geq \log_{1/\theta}\left(\frac{2\|\boldsymbol{x}^0 - \boldsymbol{x}^*\|}{\sqrt{\epsilon}}\right).$$

Therefore, the IFO complexity is

$$\tau\left[1 + \frac{1}{\omega} + \cdots + \frac{1}{\omega^{T-1}}\right] = \tau\frac{(1/\omega)^{\log_{1/\theta}\left(\frac{2\|\boldsymbol{x}^0 - \boldsymbol{x}^*\|}{\sqrt{\epsilon}}\right)} - 1}{1/\omega - 1} \overset{\textcircled{1}}{=} \frac{\tau}{1/\omega - 1}\left[\frac{4\|\boldsymbol{x}^0 - \boldsymbol{x}^*\|^2}{\epsilon} - 1\right]$$

$$\overset{}{\leq} \frac{\tau}{1/\omega - 1}\left[\frac{\|\boldsymbol{x}^0 - \boldsymbol{x}^*\|^2}{\epsilon}\right] \overset{\textcircled{2}}{\leq} \frac{81B\kappa_{\widehat{s}}}{4(\sqrt{\rho_{\widehat{s}}} + \sqrt{\ell_{\widehat{s}}})^4\epsilon}\frac{1}{1/\sqrt{1 - \frac{1}{18\sqrt{\kappa_{\widehat{s}}}}} - 1} \overset{\textcircled{3}}{\leq} \frac{81B \cdot 36\sqrt{\kappa_{\widehat{s}}}}{\rho_{\widehat{s}}(\sqrt{\rho_{\widehat{s}}} + \sqrt{\ell_{\widehat{s}}})^2\epsilon}\frac{\ell_s}{(\sqrt{\rho_{\widehat{s}}} + \sqrt{\ell_{\widehat{s}}})^2}$$

$$\leq \frac{81B \cdot 36\sqrt{\kappa_{\widehat{s}}}}{\rho_{\widehat{s}}(\sqrt{\rho_{\widehat{s}}} + \sqrt{\ell_{\widehat{s}}})^2\epsilon}\frac{\ell_{\widehat{s}}}{(\sqrt{\rho_{\widehat{s}}} + \sqrt{\ell_{\widehat{s}}})^2} = \mathcal{O}\left(\frac{\sqrt{\kappa_{\widehat{s}}}B}{\rho_s\ell_{\widehat{s}}\epsilon}\right) \overset{\textcircled{4}}{=} \mathcal{O}\left(\frac{\sqrt{\kappa_{\widehat{s}}}}{\epsilon}\right)$$

where ① uses $\omega = \theta^2$; ② uses $\theta \leq 1 - \frac{1}{18\sqrt{\kappa_{\widehat{s}}}}$ and $\tau \geq \frac{81B\kappa_{\widehat{s}}}{4(\sqrt{\rho_{\widehat{s}}} + \sqrt{\ell_{\widehat{s}}})^4\|\boldsymbol{x}^0 - \boldsymbol{x}^*\|^2}$; ③ uses $\frac{1}{1/\sqrt{1 - \frac{1}{18\sqrt{\kappa_{\widehat{s}}}}} - 1} \leq \frac{1}{1 - \sqrt{1 - \frac{1}{18\sqrt{\kappa_{\widehat{s}}}}}} \leq 36\sqrt{\kappa_{\widehat{s}}}$ since we have $1 - \sqrt{1 - a} \geq \frac{1}{2}a$ for $a \in (0, 1)$; ④ holds since (1) the parameter $\rho_{\widehat{s}}$ is the strong convex parameter at sparsity/low-rank level $s$ and thus is not very small since $\widehat{s}$ is much smaller than the feature dimension, (2) we can always scale the problem such that $\rho_{\widehat{s}}$ is not small. Notice, such a scale does not affect the ratio $B/\ell_{\widehat{s}}$ since they always scale at the same order. Thus, we have the IFO complexity $\mathcal{O}\left(\frac{\sqrt{\kappa_{\widehat{s}}}}{\epsilon}\right)$.

On the other hand, we have

$$\log_{1/\theta}\left(\frac{1}{\sqrt{\epsilon}}\right) = \frac{\log\left(\frac{1}{\sqrt{\epsilon}}\right)}{\log\left(\frac{1}{\theta}\right)} = \frac{\log\left(\frac{1}{\sqrt{\epsilon}}\right)}{\log\left(1 + \frac{1}{18\sqrt{\kappa_{\widehat{s}}} - 1}\right)} = \frac{\log\left(\frac{1}{\sqrt{\epsilon}}\right)}{\log\left(1 + \frac{1}{18\sqrt{\kappa_{\widehat{s}} - 1}}\right)} \overset{\textcircled{1}}{\leq} \mathcal{O}\left(\sqrt{\kappa_{\widehat{s}}}\log\left(\frac{1}{\sqrt{\epsilon}}\right)\right),$$

where holds since we have $\log(1 + x) \geq \log(2) \cdot x$ for $x \in [0, 1]$. The proof is completed. □

## D Proof of Auxiliary Lemmas

### D.1 Proof of Lemma 4

*Proof.* Firstly, for both vector $\boldsymbol{x}$ and matrix variable $\boldsymbol{x}$ we can decompose $\mathbb{E}\|\boldsymbol{g}^t\|^2$ and bound it as follows:

$$\mathbb{E}\|\boldsymbol{g}_\mathcal{I}^t\|^2 = \mathbb{E}\|\boldsymbol{g}_\mathcal{I}^t - \nabla_\mathcal{I}f(\boldsymbol{x}^t) + \nabla_\mathcal{I}f(\boldsymbol{x}^t) - \nabla_\mathcal{I}f(\boldsymbol{x}^*) + \nabla_\mathcal{I}f(\boldsymbol{x}^*)\|^2$$

$$\leq 3\mathbb{E}\|\boldsymbol{g}_\mathcal{I}^t - \nabla_\mathcal{I}f(\boldsymbol{x}^t)\|^2 + 3\mathbb{E}\|\nabla_\mathcal{I}f(\boldsymbol{x}^t) - \nabla_\mathcal{I}f(\boldsymbol{x}^*)\|_2^2 + 3\|\nabla_\mathcal{I}f(\boldsymbol{x}^*)\|^2$$

$$\overset{\textcircled{1}}{\leq} \frac{3}{s_t}B_t + 3\mathbb{E}\|\nabla_\mathcal{I}f(\boldsymbol{x}^t) - \nabla_\mathcal{I}f(\boldsymbol{x}^*)\|^2 + 3\|\nabla_\mathcal{I}f(\boldsymbol{x}^*)\|^2,$$

where ① use Lemma 3. Now we bound the second term. We define a function

$$h_i(\boldsymbol{x}) = f_i(\boldsymbol{x}) - f_i(\boldsymbol{x}^*) - \langle \nabla_{\mathcal{I}} f_i(\boldsymbol{x}^*), \boldsymbol{x} - \boldsymbol{x}^* \rangle.$$

It is easy to check that $\nabla h_i(\boldsymbol{x}^*) = 0$, which implies $h_i(\boldsymbol{x}^*) = \min_{\boldsymbol{x}} h_i(\boldsymbol{x})$. In this way, for vector variable $\boldsymbol{x}$, we have

$$
\begin{aligned}
0 = h_i(\boldsymbol{x}^*) \leq \min_{\eta} h_i(\boldsymbol{x} - \eta\nabla_{\mathcal{I}}h_i(\boldsymbol{x})) &\leq \min_{\eta} h_i(\boldsymbol{x}) - \eta\langle\nabla h_i(\boldsymbol{x}), \nabla_{\mathcal{I}}h_i(\boldsymbol{x})\rangle + \frac{\eta^2 \ell_s}{2}\|\nabla_{\mathcal{I}}h_i(\boldsymbol{x})\|_2^2 \\
&\overset{①}{=} \min_{\eta} h_i(\boldsymbol{x}) - \eta\|\nabla_{\mathcal{I}}h_i(\boldsymbol{x})\|^2 + \frac{\eta^2 \ell_s}{2}\|\nabla_{\mathcal{I}}h_i(\boldsymbol{x})\|^2 \\
&\overset{②}{=} h_i(\boldsymbol{x}) - \frac{1}{2\ell_s}\|\nabla_{\mathcal{I}}h_i(\boldsymbol{x})\|_2^2,
\end{aligned}
$$
(24)

where ① holds since for vector $\boldsymbol{x}$, we have $\langle\nabla h_i(\boldsymbol{x}), \nabla_{\mathcal{I}}h_i(\boldsymbol{x})\rangle = \|\nabla_{\mathcal{I}}h_i(\boldsymbol{x})\|^2$ and ② holds by optimizing $\eta = \frac{1}{\ell_s}$.

Now we consider the matrix variable $\boldsymbol{x}$. Firstly, assume that the skinny SVDs of $\boldsymbol{x}^t$, $\boldsymbol{x}^{t+1}$, $\boldsymbol{x}^*$ are respectively $\boldsymbol{x}^t = \boldsymbol{U}_t\boldsymbol{\Sigma}_t\boldsymbol{V}_t^T$, $\boldsymbol{x}^{t+1} = \boldsymbol{U}_{t+1}\boldsymbol{\Sigma}_{t+1}\boldsymbol{V}_{t+1}^T$ and $\boldsymbol{x}^* = \boldsymbol{U}_*\boldsymbol{\Sigma}_*\boldsymbol{V}_*^T$. Then we have $\boldsymbol{x}^t = \boldsymbol{U}_t\boldsymbol{U}_t^T\boldsymbol{x}^t$, $\boldsymbol{x}^{t+1} = \boldsymbol{U}_{t+1}\boldsymbol{U}_{t+1}^T\boldsymbol{x}^{t+1}$ and $\boldsymbol{x}^* = \boldsymbol{U}_*\boldsymbol{U}_*^T\boldsymbol{x}^*$. We further define the projection operation $\mathcal{P}_{\boldsymbol{U}}(\boldsymbol{x}) = \boldsymbol{U}\boldsymbol{U}^T\boldsymbol{x}$ which projects $\boldsymbol{x}$ in the subspace spanned by $\boldsymbol{U}$. Then let $\boldsymbol{v} = \boldsymbol{x}^t - \eta\boldsymbol{g}_{\mathcal{I}}^t$ and $\mathcal{I} = \mathcal{I}^* \cup \mathcal{I}^t \cup \mathcal{I}^{t+1}$, where $\mathcal{I}^* = \mathsf{supp}(\boldsymbol{x}^*)$, $\mathcal{I}^t = \mathsf{supp}(\boldsymbol{x}^t)$ and $\mathcal{I}^{t+1} = \mathsf{supp}(\boldsymbol{x}^{t+1})$. The notation $\boldsymbol{x}_{\mathcal{I}^t}^t = \boldsymbol{U}_t\boldsymbol{U}_t^T\boldsymbol{x}^t = \mathcal{P}_{\boldsymbol{U}^t}(\boldsymbol{x}^t)$. For the specifical notation $\boldsymbol{g}_{\mathcal{I}}^t$, it denotes $\boldsymbol{g}_{\mathcal{I}}^t = \mathcal{P}_{\mathcal{I}}(\boldsymbol{g}^t) = (\boldsymbol{U}_t\boldsymbol{U}_t^T + \boldsymbol{U}_{t+1}\boldsymbol{U}_{t+1}^T + \boldsymbol{U}_*\boldsymbol{U}_*^T - \boldsymbol{U}_t\boldsymbol{U}_t^T\boldsymbol{U}_{t+1}\boldsymbol{U}_{t+1}^T - \boldsymbol{U}_t\boldsymbol{U}_t^T\boldsymbol{U}_*\boldsymbol{U}_*^T - \boldsymbol{U}_{t+1}\boldsymbol{U}_{t+1}^T\boldsymbol{U}_*\boldsymbol{U}_*^T + \boldsymbol{U}_t\boldsymbol{U}_t^T\boldsymbol{U}_{t+1}\boldsymbol{U}_{t+1}^T\boldsymbol{U}_*\boldsymbol{U}_*^T)\boldsymbol{g}^t$ where $\mathcal{P}_{\mathcal{I}}$ is a projection. Then we also have

$$\langle\nabla h_i(\boldsymbol{x}), \nabla_{\mathcal{I}}h_i(\boldsymbol{x})\rangle = \langle\nabla h_i(\boldsymbol{x}), \mathcal{P}_{\mathcal{I}}(\nabla h_i(\boldsymbol{x}))\rangle \overset{①}{=} \langle\mathcal{P}_{\mathcal{I}}(\nabla h_i(\boldsymbol{x})), \mathcal{P}_{\mathcal{I}}(\nabla h_i(\boldsymbol{x}))\rangle = \|\nabla_{\mathcal{I}}h_i(\boldsymbol{x})\|^2,$$

where ① holds since $\langle\mathcal{P}_{\mathcal{I}}(\nabla_{\mathcal{I}}h_i(\boldsymbol{x})), \mathcal{P}_{\mathcal{I}}(\nabla_{\mathcal{I}}h_i(\boldsymbol{x}))\rangle = \langle\nabla h_i(\boldsymbol{x}), \mathcal{P}_{\mathcal{I}}^T\mathcal{P}_{\mathcal{I}}(\nabla_{\mathcal{I}}h_i(\boldsymbol{x}))\rangle = \langle\nabla h_i(\boldsymbol{x}), \mathcal{P}_{\mathcal{I}}(\nabla_{\mathcal{I}}h_i(\boldsymbol{x}))\rangle$ due to $\mathcal{P}_{\mathcal{I}}^T\mathcal{P}_{\mathcal{I}} = \mathcal{P}_{\mathcal{I}}$.

Thus, Eqn. (24) holds for both vector variable $\boldsymbol{x}$ and matrix variable $\boldsymbol{x}$. It further yields

$$\|\nabla_{\mathcal{I}}f_i(\boldsymbol{x}) - \nabla_{\mathcal{I}}f_i(\boldsymbol{x}^*)\|_2^2 \leq 2\ell_s\left(f_i(\boldsymbol{x}) - f_i(\boldsymbol{x}^*) - \langle\nabla_{\mathcal{I}}f_i(\boldsymbol{x}^*), \boldsymbol{x} - \boldsymbol{x}^*\rangle\right).$$

Then we are ready to bound the second term:

$$
\begin{aligned}
\mathbb{E}\|\nabla_{\mathcal{I}}f(\boldsymbol{x}^t) - \nabla_{\mathcal{I}}f(\boldsymbol{x}^*)\|^2 &= \frac{1}{n}\sum_{i=1}^n \|\nabla_{\mathcal{I}}f_i(\boldsymbol{x}^t) - \nabla_{\mathcal{I}}f_i(\boldsymbol{x}^*)\|^2 \\
&\leq \frac{1}{n}\sum_{i=1}^n 2\ell_s\left(f_i(\boldsymbol{x}^t) - f_i(\boldsymbol{x}^*) - \langle\nabla_{\mathcal{I}}f_i(\boldsymbol{x}^*), \boldsymbol{x}^t - \boldsymbol{x}^*\rangle\right) \\
&= 2\ell_s\left(f(\boldsymbol{x}^t) - f(\boldsymbol{x}^*) + \langle\nabla_{\mathcal{I}}f(\boldsymbol{x}^*), \boldsymbol{x}^t - \boldsymbol{x}^*\rangle\right).
\end{aligned}
$$

Therefore, for both vector variable $\boldsymbol{x}$ and matrix variable $\boldsymbol{x}$ we have

$$\mathbb{E}\|\boldsymbol{g}_{\mathcal{I}}^t\|^2 \leq \frac{3}{s_t}B_t + 6\ell_s\left(f(\boldsymbol{x}^t) - f(\boldsymbol{x}^*) + \langle\nabla_{\mathcal{I}}f(\boldsymbol{x}^*), \boldsymbol{x}^t - \boldsymbol{x}^*\rangle\right) + 3\|\nabla_{\mathcal{I}}f(\boldsymbol{x}^*)\|^2.$$

This completes the proof. $\qquad\square$

### D.2  Proof of Lemma 5

*Proof.* We first consider vector variable $\boldsymbol{x}$. From the strong convexity we get

$$
\begin{aligned}
f(\boldsymbol{x}^*) &\geq f(\boldsymbol{x}^t) + \langle\nabla f(\boldsymbol{x}^t), \boldsymbol{x}^* - \boldsymbol{x}^t\rangle + \frac{\rho_s}{2}\|\boldsymbol{x}^* - \boldsymbol{x}^t\|^2 \\
&\overset{①}{=} f(\boldsymbol{x}^t) + \langle\nabla_{\mathcal{I}^t\cup\mathcal{I}^*}f(\boldsymbol{x}^t) - \boldsymbol{g}_{\mathcal{I}^t\cup\mathcal{I}^*}^t + \boldsymbol{g}_{\mathcal{I}^t\cup\mathcal{I}^*}^t, \boldsymbol{x}^* - \boldsymbol{x}^t\rangle + \frac{\rho_s}{2}\|\boldsymbol{x}^* - \boldsymbol{x}^t\|^2 \\
&\overset{②}{\geq} f(\boldsymbol{x}^t) - \frac{2}{\rho_s}\|\nabla f(\boldsymbol{x}^t) - \boldsymbol{g}^t\|^2 - \frac{2}{\rho_s}\|\boldsymbol{g}_{\mathcal{I}^t\cup\mathcal{I}^*}^t\|^2 - \frac{\rho_s}{4}\|\boldsymbol{x}^* - \boldsymbol{x}^t\|^2 + \frac{\rho_s}{2}\|\boldsymbol{x}^* - \boldsymbol{x}^t\|^2 \\
&= f(\boldsymbol{x}^t) - \frac{2}{\rho_s}\|f(\boldsymbol{x}^t) - \boldsymbol{g}^t\|^2 - \frac{2}{\rho_s}\|\boldsymbol{g}_{\mathcal{I}^t\cup\mathcal{I}^*}^t\|^2 + \frac{\rho_s}{4}\|\boldsymbol{x}^* - \boldsymbol{x}^t\|^2,
\end{aligned}
$$
(25)

where ② holds since we use $\langle \boldsymbol{x}, \boldsymbol{y} \rangle \geq -\left(\frac{1}{2c}\|\boldsymbol{x}\|_2^2 + \frac{c}{2}\|\boldsymbol{y}\|_2^2\right)$ for arbitrary $c \geq 0$. By rearranging both sides of the above we get the desired bound.

Then we consider the matrix variable $\boldsymbol{x} \in \mathbb{R}^{d_1 \times d_2}$ for rank-constrained problem. Firstly, assume that the skinny SVDs of $\boldsymbol{x}^t$ and $\boldsymbol{x}^*$ are respectively $\boldsymbol{x}^t = \boldsymbol{U}_t \boldsymbol{\Sigma}_t \boldsymbol{V}_t^T$ and $\boldsymbol{x}^* = \boldsymbol{U}_* \boldsymbol{\Sigma}_* \boldsymbol{V}_*^T$. Then we have $\boldsymbol{x}^t = \boldsymbol{U}_t \boldsymbol{U}_t^T \boldsymbol{x}^t$ and $\boldsymbol{x}^* = \boldsymbol{U}_* \boldsymbol{U}_*^T \boldsymbol{x}^*$. We further define the projection operation $\mathcal{P}_{\boldsymbol{U}}(\boldsymbol{x}) = \boldsymbol{U}\boldsymbol{U}^T\boldsymbol{x}$ which projects $\boldsymbol{x}$ in the subspace spanned by $\boldsymbol{U}$. Then let $\mathcal{I} = \mathcal{I}^* \cup \mathcal{I}^t$, where $\mathcal{I}^* = \mathsf{supp}(\boldsymbol{x}^*)$ and $\mathcal{I}^t = \mathsf{supp}(\boldsymbol{x}^t)$. The notation $\boldsymbol{x}_{\mathcal{I}^t}^t = \boldsymbol{U}_t \boldsymbol{U}_t^T \boldsymbol{x}^t = \mathcal{P}_{\boldsymbol{U}^t}(\boldsymbol{x}^t)$. For the specifical notation $\nabla_{\mathcal{I}} f(\boldsymbol{x}^t)$, it denotes $\nabla_{\mathcal{I}} f(\boldsymbol{x}^t) = \mathcal{P}_{\mathcal{I}}(\nabla f(\boldsymbol{x}^t)) = (\boldsymbol{U}_t \boldsymbol{U}_t^T + \boldsymbol{U}_* \boldsymbol{U}_*^T - \boldsymbol{U}_t \boldsymbol{U}_t^T \boldsymbol{U}_* \boldsymbol{U}_*^T)\nabla f(\boldsymbol{x}^t)$ where $\mathcal{P}_{\mathcal{I}}$ is a projection. Then we have

$$\mathcal{P}_{\mathcal{I}}(\boldsymbol{x}^* - \boldsymbol{x}^t) = (\boldsymbol{U}_t \boldsymbol{U}_t^T + \boldsymbol{U}_* \boldsymbol{U}_*^T - \boldsymbol{U}_t \boldsymbol{U}_t^T \boldsymbol{U}_* \boldsymbol{U}_*^T)(\boldsymbol{U}_* \boldsymbol{U}_*^T \boldsymbol{x}^* - \boldsymbol{U}_t \boldsymbol{U}_t^T \boldsymbol{x}^t) = \boldsymbol{x}^* - \boldsymbol{x}^t,$$

which further gives

$$\langle \nabla f(\boldsymbol{x}^t), \boldsymbol{x}^* - \boldsymbol{x}^t \rangle = \langle \nabla f(\boldsymbol{x}^t), \mathcal{P}_{\mathcal{I}}(\boldsymbol{x}^* - \boldsymbol{x}^t) \rangle \overset{①}{=} \langle \mathcal{P}_{\mathcal{I}}(\nabla f(\boldsymbol{x}^t)), \mathcal{P}_{\mathcal{I}}(\boldsymbol{x}^* - \boldsymbol{x}^t) \rangle$$
$$= \langle \nabla_{\mathcal{I}} f(\boldsymbol{x}^t), \boldsymbol{x}^* - \boldsymbol{x}^t \rangle,$$

where ① holds since $\langle \mathcal{P}_{\mathcal{I}}(\nabla f(\boldsymbol{x}^t)), \mathcal{P}_{\mathcal{I}}(\boldsymbol{x}^* - \boldsymbol{x}^t) \rangle = \langle \nabla f(\boldsymbol{x}^t), \mathcal{P}_{\mathcal{I}}^T \mathcal{P}_{\mathcal{I}}(\boldsymbol{x}^* - \boldsymbol{x}^t) \rangle = \langle \nabla f(\boldsymbol{x}^t), \mathcal{P}_{\mathcal{I}}(\boldsymbol{x}^* - \boldsymbol{x}^t) \rangle$ due to $\mathcal{P}_{\mathcal{I}}^T \mathcal{P}_{\mathcal{I}} = \mathcal{P}_{\mathcal{I}}$.

In this way, ① in Eqn. (25) holds. Thus, the above result also hold for matrix variable in rank-constrained problem. The proof is completed. $\square$

### D.3   Proof of Lemma 6

*Proof.* Let $\lambda_1 \leq \lambda_2 \leq \cdots \leq \lambda_d$ be the eigenvalues of $\boldsymbol{A}$ and $\boldsymbol{\Lambda}$ be a diagonal matrix whose diagonal entries are $\{\lambda_i\}$ in a non-decreasing order. By proper manipulation we get

$$\rho\left(\begin{bmatrix} (1+\nu)\boldsymbol{I} - \eta\boldsymbol{A} & -\nu\boldsymbol{I} \\ \boldsymbol{I} & 0 \end{bmatrix}\right) = \rho\left(\begin{bmatrix} (1+\nu)\boldsymbol{I} - \eta\boldsymbol{\Lambda} & -\nu\boldsymbol{I} \\ \boldsymbol{I} & 0 \end{bmatrix}\right) = \max_{i \in [d]} \rho\left(\begin{bmatrix} 1+\nu - \eta\lambda_i & -\nu \\ 1 & 0 \end{bmatrix}\right),$$

where in the second equality we have used the fact that it is possible to permute the matrix to a block diagonal matrix with $2 \times 2$ blocks. For each $i \in [d]$, the eigenvalues of the $2 \times 2$ matrices are given by the roots of

$$\lambda^2 - (1 + \nu - \eta\lambda_i)\lambda + \nu = 0.$$

Given that $\nu \geq |1 - \sqrt{\eta\lambda_i}|^2$, the roots of the above equation are imaginary and both have magnitude $\sqrt{\nu}$. Since $\nu = \max\{|1 - \sqrt{\eta\mu}|^2, |1 - \sqrt{\eta\ell}|^2\}$, the magnitude of each root is at most $\max\{|1 - \sqrt{\eta\mu}|, |1 - \sqrt{\eta\ell}|\}$. This proves the desired spectral norm bound. $\square$

## E   Additional Experimental Results

### E.1   Descriptions of Testing Datasets

We briefly introduce the seven testing datasets in the manuscript. Among them, three datasets are provided in the LibSVM website[1], including rcv1, real-sim and epsilon. We also evaluate our algorithms on mnist[2] for handwritting recognition, news20[3] for news classification, coil100[4] and caltech256[5] for image classification. Their detailed information is summarized in Table 2. We can observe that these datasets are different from each other in feature dimension, training samples, and class numbers, *etc*. It should be mentioned that for caltech256 including 256 kinds of objects and one background class, we use its OverFeat feature, while for other datasets, we all use their raw data.

Table 2: Descriptions of the ten testing datasets.

|  | #class | #sample | #feature |  | #class | #sample | #feature |
|---|---|---|---|---|---|---|---|
| rcv1 | 2 | 20,242 | 47,236 | news20 | 20 | 62,061 | 15,935 |
| real-sim | 2 | 72,309 | 20,958 | coil100 | 100 | 7,200 | 1,024 |
| epsilon | 2 | 100,000 | 2,000 | caltech256 | 257 | 5,140 | 2,000 |
| mnist | 10 | 60,000 | 784 |  |  |  |  |

## E.2 More Experiments on A Single Pass over Data

Finally, we give more experimental results on a single pass over data. Following the setting in Section 5 in manuscript, here we test the considered algorithms on logistic regression with regularization parameters $\lambda = 10^{-5}$. We follow our theoretical results to exponentially expand the mini-batch size $s_k$ in HSG-HT and AHSG-HT and set $\tau = 1$. Figure 3 summarizes the numerical results in this setting. One can observe that on these optimization problems, most algorithms still achieve high accuracy after one pass over data, while HSG-HT and AHSG-HT also converge significantly faster than the other algorithms. These observations are consistent with the results in Figure 1 in the manuscript. All these results demonstrate the high efficiency of HSG-HT and AHSG-HT and also confirm the theoretical implication of Corollary 1 and 2 that HSG-HT and AHSG-HT always have lowest hard thresholding complexity than the compared algorithms and have lower in IFO complexity than other considered variance-reduced algorithms linearly depending on the sample size $n$, when the desired accuracy is moderately small and data scale is large.

Figure 3: Single-epoch processing: comparison among hard thresholding algorithms for a single pass over data on sparse logistic regression with regularization parameter $\lambda = 10^{-5}$.

Since the objective loss decreases fast along with the hard thresholding iteration, here we magnify the subfigures in Figure 1 in the manuscript and the above Figure 3 which display the objective loss decrease along with the hard thresholding iteration. In this way, the objective loss decrease along with the hard thresholding iteration can be viewed better. From Figure 4, one can easily observe that AHSG-HT and HSG-HT converge much faster than the compared algorithms. Moreover, AHSG-HT achieves higher optimization accuracy which demonstrates that AHSG-HT is superior over HSG-HT in hard thresholding complexity. All these results confirm the our theoretical implication of Corollary 1 and 2.

Figure 4: Comparison of hard thresholding complexity in single-epoch processing. (a) magnifies the hard thresholding iterations in Figure 1 in the manuscript. (b) magnifies the hard thresholding iterations in Figure 3 above.

## Footnotes

[1]https://www.csie.ntu.edu.tw/ cjlin/libsvmtools/datasets/

[2]http://yann.lecun.com/exdb/mnist/

[3]http://qwone.com/ jason/20Newsgroups/

[4]http://www1.cs.columbia.edu/CAVE/software/softlib/coil-100.php

[5]https://authors.library.caltech.edu/7694/