[Reviews · NeurIPS 2018]

Reviewer 1



The paper deals with the convergence properties of stochastic incremental hybrid gradient hard thresholding methods. It proves sample-size-independent gradient evaluation and hard thresholding complexity bounds similar to full gradient methods.As the per-iteration hard thresholding can be as expensive as SVD in rank-constrained problems, it is important for iterations to converge quickly while using a minimal number of hard thresholding operations. The paper gives a rigorous analysis of convergence in "big data" settings and also shows the heavy ball accelerations also yield expected benefits.

Reviewer 2



The article analyses convergence in hard-thresholding algorithms and proposes an accelerated stochastic hybrid hard thresholding method that displays better convergence with respect to the compared methods. The article is dense but relatively fine to follow. Theoretical development seems to be complete and accurate, though I admit I have not throughly followed the full derivation. Experimental section is in accordance with the theoretical claims and is more than sufficient. Just for the sake of reproducibility of the results an exhaustive pseudocode or repository should be made available as a companion to the article to further strength the autor's points.

Reviewer 3



This paper proposes to use HSGD with hard thresholding for L0 or rank constrained problems. The paper exhibits improved rates compared to state-of-the-art SVRG and SGD approaches, and proposes an accelerated variant using the heavy-ball method. Numerical experiments are convincing and illustrate the improvement compared to state-of-the-art baselines. I think that this is a good paper, which provides an interesting contribution to the field.